# Deep transfer learning of cancer drug responses by integrating bulk and single-cell RNA-seq data

Junyi Chen[1,6], Xiaoying Wang[2,6], Anjun Ma ©[1,3] ✉, Qi-En Wang[4], Bingqiang Liu[2], Lang Li[1], Dong Xu ©[5] & Qin Ma ©[1,3] ✉

Drug screening data from massive bulk gene expression databases can be analyzed to determine the optimal clinical application of cancer drugs. The growing amount of single-cell RNA sequencing (scRNA-seq) data also provides insights into improving therapeutic effectiveness by helping to study the heterogeneity of drug responses for cancer cell subpopulations. Developing computational approaches to predict and interpret cancer drug response in single-cell data collected from clinical samples can be very useful. We propose scDEAL, a deep transfer learning framework for cancer drug response prediction at the single-cell level by integrating large-scale bulk cell-line data. The highlight in scDEAL involves harmonizing drug-related bulk RNA-seq data with scRNA-seq data and transferring the model trained on bulk RNA-seq data to predict drug responses in scRNA-seq. Another feature of scDEAL is the integrated gradient feature interpretation to infer the signature genes of drug resistance mechanisms. We benchmark scDEAL on six scRNA-seq datasets and demonstrate its model interpretability via three case studies focusing on drug response label prediction, gene signature identification, and pseudotime analysis. We believe that scDEAL could help study cell reprogramming, drug selection, and repurposing for improving therapeutic efficacy.

Precision medicine has achieved remarkable success in understanding the complexity of the genomic landscape of cancer. The idea of tailoring cancer treatments to the particular genomic signature of an individual cell is gaining traction. Several in vitro drug screening studies have been conducted, giving rise to drug response data on various cancer cell lines[1,2]. However, cancer drug treatments suffer from low efficacies and high relapse rates caused by cancer heterogeneity among diverse states or cell fates. Such heterogeneity is responsible for differentiated responses of individual cells to a drug, leading to a minimal amount of cancerous residues remaining in the body, followed by, ultimately, cancer relapse[3]. The single-cell RNA-sequencing

(scRNA-seq) technique provides an unprecedented opportunity to discover the heterogeneous gene expressions of cancer subpopulations in response to specific drugs[4]. Existing drug-response prediction methods developed for bulk data cannot be directly used for larger-scale and highly intricate single-cell data. Hence, computational methods to infer cancer drug responses at the single-cell level are urgently needed.

Deep learning methods have been deployed to tackle scRNA-seq data, redefining our capabilities to analyze large-scale data using sophisticated architectures of artificial neural networks. Deep learning models applied to scRNA-seq data have achieved competitive

[1]Department of Biomedical Informatics, College of Medicine, The Ohio State University, Columbus, OH 43210, USA. [2]Department of Mathematics, Shandong University, Shandong 250100, China. [3]Pelotonia Institute for Immuno-Oncology, The James Comprehensive Cancer Center, The Ohio State University, Columbus, OH 43210, USA. [4]Department of Radiation Oncology, Comprehensive Cancer Center, The Ohio State University, Columbus, OH 43210, USA. [5]Department of Electrical Engineering and Computer Science, and Christopher S. Bond Life Sciences Center, University of Missouri, Columbia, MO 65211, USA. [6]These authors contributed equally: Junyi Chen, Xiaoying Wang. ✉e-mail: anjun.ma@osumc.edu; qin.ma@osumc.edu

performances in gene expression imputation, cell clustering, batch correction, and similar tasks[5–7]. The main obstacle in developing a deep learning-based tool for predicting single-cell drug responses is insufficient training power owing to the limited number of benchmarked data in the public domain. Intuitively, drug-related bulk RNA-seq data can be effective complementary resources to infer gene expression-drug response relations in support of the drug response predictions at the single-cell level[8,9]. Fortunately, deep transfer learning (DTL) can transfer knowledge and relation patterns from bulk data to single-cell data, which can be a means to overcome the issue of limited training data[10]. Using transfer learning, we can solve a particular task at the single-cell level using a preliminary trained model at the bulk level, either entirely or partially. The DTL model has been applied as an effective strategy in leveraging multiple bulk data sources for cancer drug response predictions[11]; however, thus far, its capabilities in transferring valuable bulk-level knowledge to the single-cell level are under-investigated.

In this work, we develop scDEAL (single-cell Drug rEsponse Ana-Lysis) by adapting a Domain-adaptive Neural Network (DaNN)[12] to predict drug responses from bulk and scRNA-seq data. scDEAL is very powerful at predicting single-cell level drug sensitivity as it establishes bridges among drug sensitivity, gene features in single cells, and gene features in bulk samples. scDEAL highlights the following aspects: (*i*) it can use a large amount of bulk-level drug response RNA-seq information from the Genomics of Drug Sensitivity in Cancer (GDSC) database[13,14] and Cancer Cell Line Encyclopedia (CCLE)[15] to train and optimize the model[16,17]; (*ii*) in order to account for data-structure differences between bulk and scRNA-seq data, scDEAL harmonizes single-cell and bulk embeddings to ensure that the drug response labels are transferable from bulk to single cells; (*iii*) in order to avoid losing heterogeneity in scRNA-seq data, scDEAL includes cell cluster labels for loss function regularization in each training epoch; (*iv*) scDEAL's integrated gradient interpretation infers the signature genes of drug response predictions, which improves the interpretability of the model. We conduct comprehensive analyses and evaluations on six benchmark drug-treated scRNA-seq data[18–22]; scDEAL achieves high accuracy in predicting cell-type drug responses. We further identify gene signatures that are considered to directly contribute to drug sensitivity or resistance in a cell by tracing and accumulating the integrated gradients of each neuron in the DTL model. Finally, we prove that the predicted drug response aligned well with the expression trajectory of treatment procedures. Overall, we believe scDEAL enables the deployment of the DTL model in single-cell drug response prediction, which may benefit preliminary studies in drug development, repurposing, and selection in cancer treatment.

## Results

### Overview of the scDEAL framework
First, scDEAL models relations between the gene expression feature and drug response at the bulk level, where annotations for cell lines are available. Then, the shared low-dimensional feature space between single-cell and bulk data is identified in order to harmonize the relation between the two data types. The *gene expression–drug response* relations at the bulk level are captured via the shared low-dimensional feature space. A DTL model is trained to learn the optimized solution to the aforementioned two relations. Finally, the *single cell–drug response* relations can be built through the meta-relation of *gene expression at the single-cell level, gene expression at the bulk level,* and *drug response* in the DTL model. Overall, scDEAL infers drug responses for individual cells without requiring supervised training at the single-cell level (Fig. 1a).

Bulk and scRNA-seq data were preprocessed prior to the input of scDEAL (Supplementary Fig. S1). The scDEAL framework involves five major steps: (1) extracting bulk gene features, (2) predicting drug response in each bulk cell line using features extracted in step 1, (3)

extracting single-cell gene features, (4) jointly training and updating all the models in the previous steps, and (5) transferring and applying the trained model to scRNA-seq data to predict drug responses (Fig. 1b). The training of scDEAL is composed of a source model for initial parameter determination of bulk-level feature reduction and drug response prediction using bulk data only, as well as a targeted model to include scRNA-seq data and deploy the transfer learning strategy to train and update the entire framework for single-cell drug response prediction. Two denoising autoencoders (DAEs) are trained to extract low-dimensional gene features from bulk and scRNA-seq data separately. The training reduces the reconstruction loss between the decoder output and the expression profiles, making low-dimensional features informative enough to represent the original gene expressions. The preliminary training is used to generate the initial neuron weights within the DTL model. A fully connected predictor is attached to the trained bulk feature extractor for predicting bulk-level drug responses.

Finally, the DTL model updates the two DAE models and the predictor model simultaneously in a multi-task learning manner. Specifically, the first task is to minimize the differences (i.e., mean maximum discrepancy loss) between gene features from two extractors, bridging the communication between bulk and scRNA-seq data. The second task is to minimize the difference between the prediction results and the database-provided drug responses via the cross-entropy loss. We expect the framework to be updated to harmonize bulk expression data and scRNA-seq data as well as to transfer the trustworthy gene-drug relations from the bulk level to the single-cell level. The output of scDEAL is the predicted potential drug response of individual cells.

One of the critical challenges in model training is maintaining single-cell heterogeneity when harmonizing scRNA-seq data with bulk data. Two strategies were applied. First, as the noise characteristics in bulk RNA-seq and scRNA-seq data are quite distinct, we used a DAE model, rather than a common autoencoder or a variational auto-encoder, to induce noises in bulk as well as scRNA-seq prior to the feature reduction. By this means, we could avoid the risk of imbalanced training that would only force gene expressions in scRNA-seq data close to bulk RNA-seq data. Second, we integrated cell-clustering results to regularize the overall loss function of scDEAL, so that cellular heterogeneity would be retained during the training process.

### Benchmarking single-cell drug response predictions in scDEAL
We evaluated the drug response prediction performances on six public scRNA-seq datasets treated by five drugs, i.e., Cisplatin, Gefitinib, I-BET-762, Docetaxel, and Erlotinib (Supplementary Table S1). All datasets have been provided with ground-truth drug response annotations (i.e., drug-sensitive or drug-resistant) for individual cells. A ground truth label is a binary indicator (0 indicates resistant and 1 indicates sensitive) extracted from the original manuscripts. Most studies determine the drug response to an entire cell group based on treatment conditions, e.g., dimethyl sulfoxide (DMSO)-treated cells are all sensitive, and cells surviving after treatment are all resistant. Compared with the ground-truth labels, scDEAL prediction was evaluated using seven metrics: F1-score, area under the receiver operating characteristic (AUROC), AP score, precision, recall, Adjusted Mutual Information (AMI), and Adjusted Rand Index (ARI). We showcase the results of F1-score, AUROC, and AP score on the six datasets (Fig. 2a) based on optimized hyperparameters in scDEAL (Supplementary Table S2), while the rest of the scores can be found in Source Data 1. The average scores of the six datasets are 0.892 (F1-score), 0.898 (AUROC), 0.944 (AP score), 0.926 (precision), 0.899 (recall), 0.528 (AMI), and 0.608 (ARI). To better visualize the prediction results, we generated UMAPs for each dataset and colored them by predicted cell clusters, ground-truth single-cell drug responses, scDEAL-predicted drug responses (binary labels as well as continuous probability), and

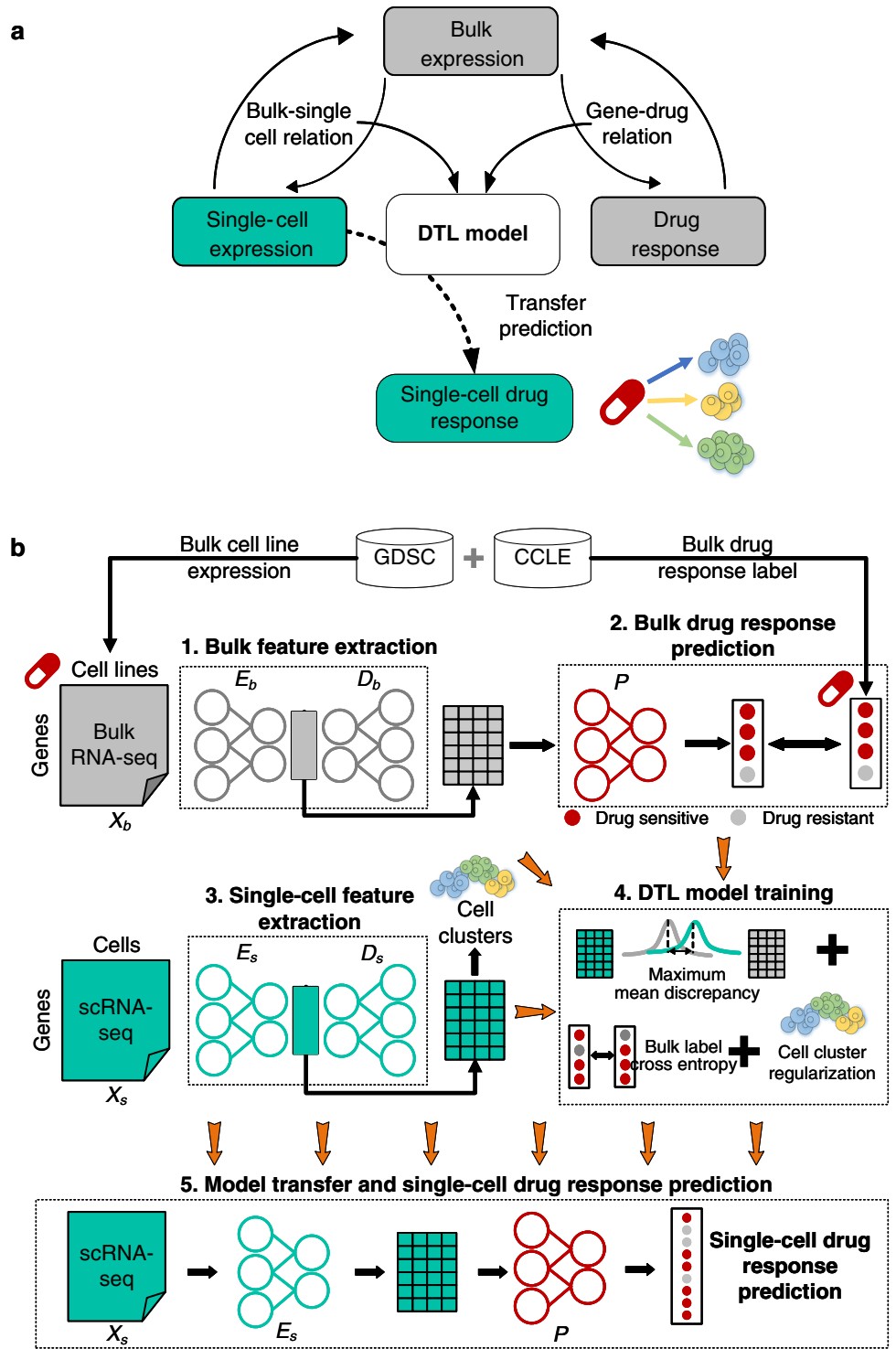

**Fig. 1 | The scDEAL framework. a** scDEAL trains the model to align two relations: (*i*) bulk−single-cell relations and (*ii*) gene−drug response relations at the bulk level. The trained model will then be transferred to be directly applied to the scRNA-seq data and to predict the single-cell drug responses. Green-colored elements represent single-cell related data, and grey-colored elements represent bulk-related data. Different colors of cells represent different cell types. **b** Bulk RNA-seq data and the corresponding drug response labels are obtained from the GDSC and CCLE databases. Five steps are then applied. A DAE is used to induce noises into the bulk data. It uses an encoder ($E_b$) and a decoder ($D_b$) to obtain low-dimensional features. The bulk feature × cell-line matrix is then input to a fully connected predictor ($P$) to predict cell-line drug responses. A similar strategy is used for single-cell feature election using a separated DAE ($E_s$ and $D_s$). The overall framework will be trained by considering the maximum mean discrepancy between the low-dimensional feature spaces of single-cell and bulk data, the cross-entropy loss between predicted bulk cell-line drug responses and ground-truth labels, and the regularization of cell clusters predicted from scRNA-seq data. By achieving the minimum overall loss, $E_b$, $E_s$, and $P$ will be updated and optimized simultaneously. scDEAL transfers the well-trained $E_s$ and $P$ to predict single-cell drug responses from the scRNA-seq data. Abbreviations: deep transfer learning (DTL), Genomics of Drug Sensitivity in Cancer (GDSC), Cancer Cell Line Encyclopedia CCLE.

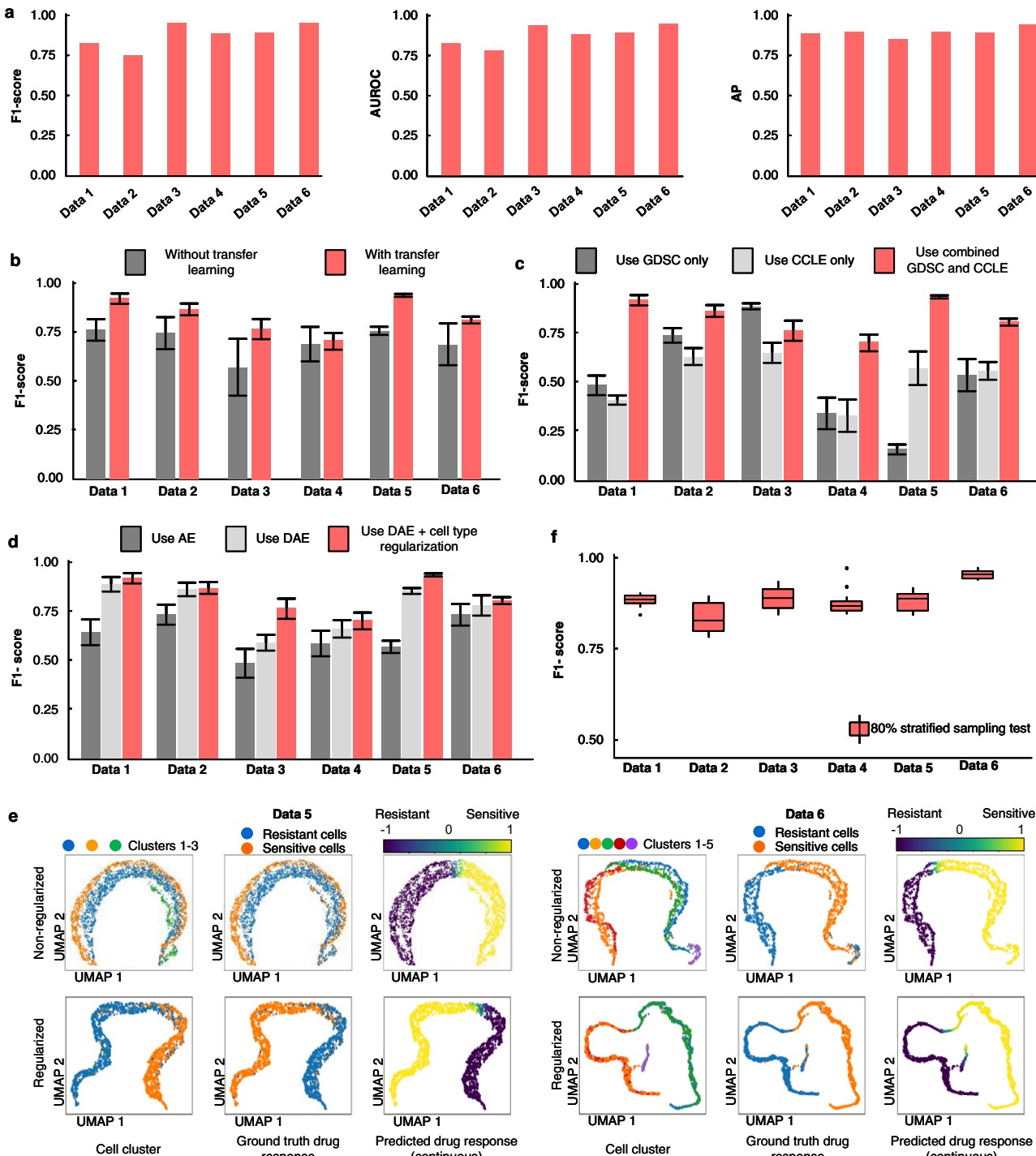

**Fig. 2 | Benchmarking results of scDEAL. a** Optimized benchmarking results of all six datasets using scDEAL. Source data are provided as Source Data 1: optimized benchmarking results of seven metrics. **b** F1-score comparison using GDSC database only, CCLE database only, and both databases in training scDEAL for all six datasets. The bar plot shows the mean F1-scores of each data (*n* = 50; same parameter settings for each data; different seeds), with error bars representing +/− standard deviations. The same rules are also applied for the bar plots in c and d. Source data are provided as Source Data 2: F1-score of 50 repeated experiments comparing with and without transfer learning in six datasets. **c** Drug response prediction comparisons of scDEAL framework using common autoencoder (dark grey), denoising autoencoder (light grey), and the combination of denoising autoencoder in feature extraction and cell-type regularization in DaNN loss function for transfer learning (pink). Source data are provided as Source Data 3: F1-score

of 50 repeated experiments comparing use GDSC, use CCLE, and use both bulk databases in six datasets. **d** Comparisons of scDEAL with (grey) and without (pink) transfer learning in terms of F1-scores. Source data are provided as Source Data 4: F1-score of 50 repeated experiments comparing use autoencoder, denoise autoencoder, and combination of denoise autoencoder and cell type regularization in six datasets. **e** Latent representations of scDEAL obtained with/without cell type regularization for Data 5 and 6. **f** Robustness test on six scRNA-seq datasets via 80% stratified sampling in terms of F1-score. Each box shows the minimum, first quartile, median, third quartile, and maximum F1-scores of 20 samplings (*n* = 20). Dots represent outliers. Source data are provided as Source Data 6: F1-score of 80% stratified sampling of 20 repeats on six datasets. Abbreviations: Genomics of Drug Sensitivity in Cancer (GDSC), Cancer Cell Line Encyclopedia (CCLE), denoising autoencoder (DAE).

generated Sankey plots to observe the discrepancies between the ground-truth and predicted labels (Supplementary Fig. S2). We observed that the predicted drug response labels of most cells were well-aligned with the ground truth and showcased distinct cell cluster differences. The prediction results at the bulk level also showed good performances, indicating that the model has been well-trained before being transferred to analyze scRNA-seq data (Supplementary Table S3 and Fig. S3).

As described above, scDEAL achieved considerably high performance in single-cell drug response prediction among all six datasets. Furthermore, to elucidate the rationale of the scDEAL framework design, we replaced or removed specific component(s) in scDEAL and compared the results with those from the final framework. The final scDEAL framework will be comprehensively validated if it can outperform all the alternative models.

First, a comparison test was performed by training the model only on the bulk data and then directly using it for scRNA-seq data prediction without step 3 (transfer learning). For each data, the experiment was repeated 50 times ($n = 50$). Noted that the result of scDEAL is fully reproducible if use the same seed for the same data training. The results on all six datasets showed a significant increase in F1-scores when using the transfer strategy compared to without it (Fig. 2b and Source Data 2). On average, scDEAL achieved a 19% increase in F1-score compared to the model without transfer learning. Our comparison showed that transfer learning contributed to the performance improvement in single-cell drug response prediction.

Second, to evaluate whether the training power of the transfer model relies on bulk resources, we benchmarked scDEAL using bulk data from the GDSC database only, CCLE database only, and a combination of GDSC and CCLE databases (Supplementary Tables S4, S5). Our results showed that combining bulk data from GDSC and CCLE databases can significantly enhance the prediction power (Fig. 2c and Source Data 3). On average, the integration of the two databases resulted in a 130% and 69% increase in the F1-scores compared to the results using only the GDSC or CCLE database, respectively.

Third, we validated whether using DAE and cell-type regularization can help reduce the loss of single-cell heterogeneity and enhance the prediction performance. We compared the results of the framework using common autoencoders for the feature extraction in bulk and scRNA-seq data, a framework using DAE but not regularized by cell type, and the final scDEAL framework (including DAE as well as cell type regularization). For all six datasets, using DAE and cell-type regularization in the framework achieved a better performance than the other two options (Fig. 2d and Source Data 4). On average, using DAE and cell type regularization showed a 36% and 9% increase in the F1-scores compared to the results only using AE or DAE database, respectively. To further elucidate how the addition of cell-type regularization can better preserve the heterogeneity of scRNA-seq data, we showcased cells with cell-cluster and drug-response annotations using latent representations from scDEAL with and without the cell-type regularizer (Fig. 2e and Supplementary Fig. S4). The UMAP results showed that, after applying the cell-type regularizer, cells become more ordered and compact within a cluster.

Furthermore, to validate whether relations between gene expression and drug responses at the bulk level have been successfully learned and transferred to the single-cell level, we calculated an integrated gradients (IG) score to reflect the potential contribution of each gene to the final drug response label prediction (Supplementary Fig. S5 and Methods). Traditional DEG analysis may lead to biased results related to cell types rather than drug response; hence, we used the differential IG scores between sensitive and resistant cells to represent genes that are critical to drug response. The IG score is based on the accumulation of gradients of neurons in a neural network following the path of layer connections. A gene with a higher IG score to the drug-sensitive labeling indicates that the gene is more related to drug sensitivity and contributes more to categorizing samples as drug-sensitive. Similar rules were applied to the resistant labels. By comparing the number of genes contributing to the drug response labeling at the bulk and single-cell levels, we found that, on average, 46% of genes were shown to be overlapped in both data types contributing to drug sensitivity, while 53% of genes were shown overlapped with drug resistance (Supplementary Table S6). The results indicate that various gene-drug relations can be inferred at the bulk and single-cell levels.

Finally, we showcased a grid parameter tuning result, including 480 combinations of six hyperparameters (e.g., bulk sampling method, predictor dimension, learning rate, single-cell encoder dimension, dropout, and bottleneck dimension) (Supplementary Fig. S6 and Source Data 5). Overall, our results showed no significant effects of single parameter selections on scDEAL performances. Four datasets, i.e., Data 1, 2, 4, and 5, are more robust on all parameter combinations than Data 3 and 6. The performance and robustness of scDEAL are likely related to the combination of parameters, but not in a sensitive fashion. For any new dataset, we recommend adjusting the bulk sampling methods and bottleneck dimensions, because we found these two parameters differed considerably among six datasets when achieving the best prediction performance (Supplementary Table S2). To evaluate the robustness of scDEAL, we performed a randomly stratified sampling test ($n = 20$) on the six datasets (Fig. 2f, Supplementary Fig. S7, and Source Data 6). The variations of F1-score, AUROC, AP score, precision, recall, AMI, and ARI are 0.031, 0.046, 0.027, 0.029, 0.031, 0.156, and 0.198, respectively, indicating that scDEAL is robust across multiple runs of random sampling.

## scDEAL achieves good drug response prediction results in leukemia cells under a variety of I-BET treatment conditions

We showcased the analytical power of scDEAL on Data 6 (GSE110894)[22], including 1419 Mixed Lineage Leukemia-AF9 (MA9) leukemic cells treated with a BET inhibitor (I-BET) (Fig. 3a). Four treatment conditions were included with two sensitive states (DMSO and I-BET 400 nM) and two resistant states (IBET-resistant and IBET-resistant withdraw)[22]. It was observed that scDEAL predicted leukemic-cell drug responses consistently as compared to the original study. We found that 97.1% of predicted drug-resistant cells and 95.8% of predicted drug-sensitive cells in scDEAL matched original labels. In addition, scDEAL provided two types of drug-response prediction scores, i.e., the continuous probability score and the binary sensitive/resistant label. A higher continuous score in a cell reflects a higher likelihood of the cell being sensitive to the drug. The binary label is determined by counting cells with a continuous probability score between 0–0.5 as resistant cells and 0.5–1 as sensitive cells.

Next, we introduce a gene score to reflect the overall gene expression level of differentially expressed genes identified in the sensitive (or resistant) cell clusters. The hypothesis behind the score is that an accurate prediction assigns the correct response label to cells. Therefore, the gene scores of DEGs between the resistant and the sensitive states for an accurate prediction should be correlated to DEGs derived from the ground truth. In addition, our DEG showed gene score patterns that can separate resistant and sensitive cells better than DEGs identified using ground-truth labels (Fig. 3b). The correlation between predicted DEG scores and ground truth DEG scores is as high as $R^2 = 0.90$ for the sensitive DEG list, and $R^2 = 0.77$ for the resistant DEG list (Fig. 3c and Source Data 7). We performed an empirical null model test to evaluate the significance of the correlation. We randomly selected the same number of genes as our predicted DEGs and calculated the correlation as described above 1000 times. Our empirical test ($n = 1000$) results showed $p$-values for bother-sensitive and resistant DEG score correlations are lower than 0.001, indicating that our correlation is significant and statistically meaningful (Fig. 3d).

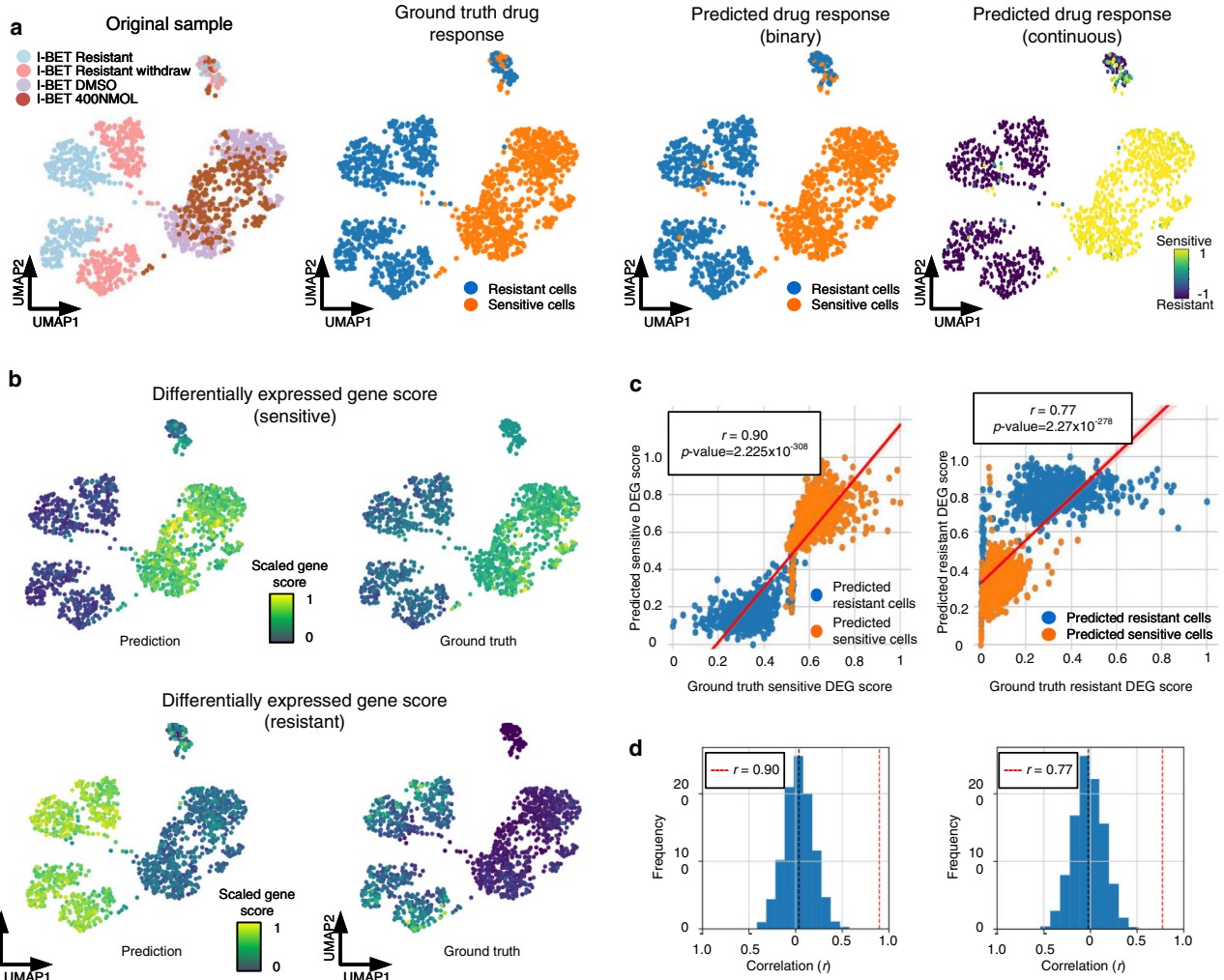

**Fig. 3 | Case study of Data 6 corresponding to I-BET treatment. a** From left to right: UMAPS visualizations of Data 6 colored by sample treatment types provided in the original study, ground-truth drug-response labels, predicted binary drug response labels, and predicted continuous drug response probability scores. **b** UMAP plot colored by sensitive (and resistant) gene scores derived from differentially expressed genes in the predicted and ground-truth sensitive (and resistant) cluster. Source data are provided as Source Data 7: sensitive and resistant DEG scores in predicted and ground truth sensitive and resistance cells in Data 6. **c** The

plot displays the one-tail Pearson's correlation test between the gene scores derived from the predicted and the ground-truth cell labels ($n = 1,404$). The error bands showed a 95% confidence interval of the regression. Source data are provided as Source Data 7: sensitive and resistant DEG scores in predicted and ground truth sensitive and resistance cells in Data 6. **d** Empirical test ($n = 1,000$) of correlation coefficient. The x-axis represents empirical correlations of differentially expressed gene scores, the y-axis represents frequencies, and the red dashed line represents scDEAL results. Abbreviation: differentially expressed gene (DEG).

## scDEAL can identify critical genes responsible for drug response

Though scDEAL delivered accurate predictions for single-cell drug responses, comprehension of the active genetic features within the model is essential. We conducted scDEAL analysis for oral squamous cell carcinoma (OSCC) treated by Cisplatin in Data 1[18]. Cisplatin exerts its anti-cancer activity via the generation of DNA crosslinks by interacting with purine bases on DNA, interfering with DNA replication and causing additional deleterious DNA double-strand breaks, which, if not repaired, can lead to apoptosis of cancer cells[23]. Thus, any factor that can enhance DNA repair or/and inhibit cellular apoptosis is able to render cancer cells resistance to Cisplatin treatment. Using scDEAL, 85% of cells were correctly predicted as either sensitive or resistant to Cisplatin, with an F1-score of 0.92, AUROC of 0.92, and AP score of 0.97 (Fig. 4a and Supplementary Fig. S8 and Source Data 8). Genes with adjusted $p$ values <0.05, log-fold change <0.1, and cell percentage in either comparison group higher than 0.2 were defined as critical genes (CGs) that impact the drug response. We identified 936 drug-sensitive CGs in the HN120P (sensitive cell group) and 868 drug-resistant CGs in the HN120PCR (resistant cell group after Cisplatin treatment over four

months) with significantly differential IG scores (Fig. 4b and Supplementary Data 1). We observed that several top predicted resistant CGs, e.g., *BCL2A1*[24] and *DKK1*[25], possess anti-apoptotic activity (Fig. 4c). Overexpression of these genes has been shown to mediate resistance to Cisplatin[26,27].

Gene Oncology (GO) pathway enrichment analysis (Supplementary Data 2) of the 868 drug-resistant CGs further revealed that the Cisplatin-resistant CGs predicted in HN120PCR cells are significantly enriched in "DNA repair" (Benjamini adjusted $p$-value = 0.039), which is one of the major Cisplatin resistance-related biological processes. Among the 26 DNA repair-related genes in the list of drug-resistant CGs in HN120PCR cells, we find strong literature evidences for eight of them *RAD51*[28], *EXO1*[29], *FANCL*[30], *MSH3*[31], *RIF1*[32], *USP28*[33], *FANCG*[34], and *POLH*[35]. These genes are critical to the DNA repair pathways that are used to deal with Cisplatin-induced DNA lesions, including inter-strand crosslinks[36] and DNA double-strand breaks[37], as well as the DNA damage tolerance pathway that is used to bypass Cisplatin-induced DNA damage to promote cancer cell survival[38]. On the other hand, another significantly enriched GO pathway, "cell division" (Benjamini

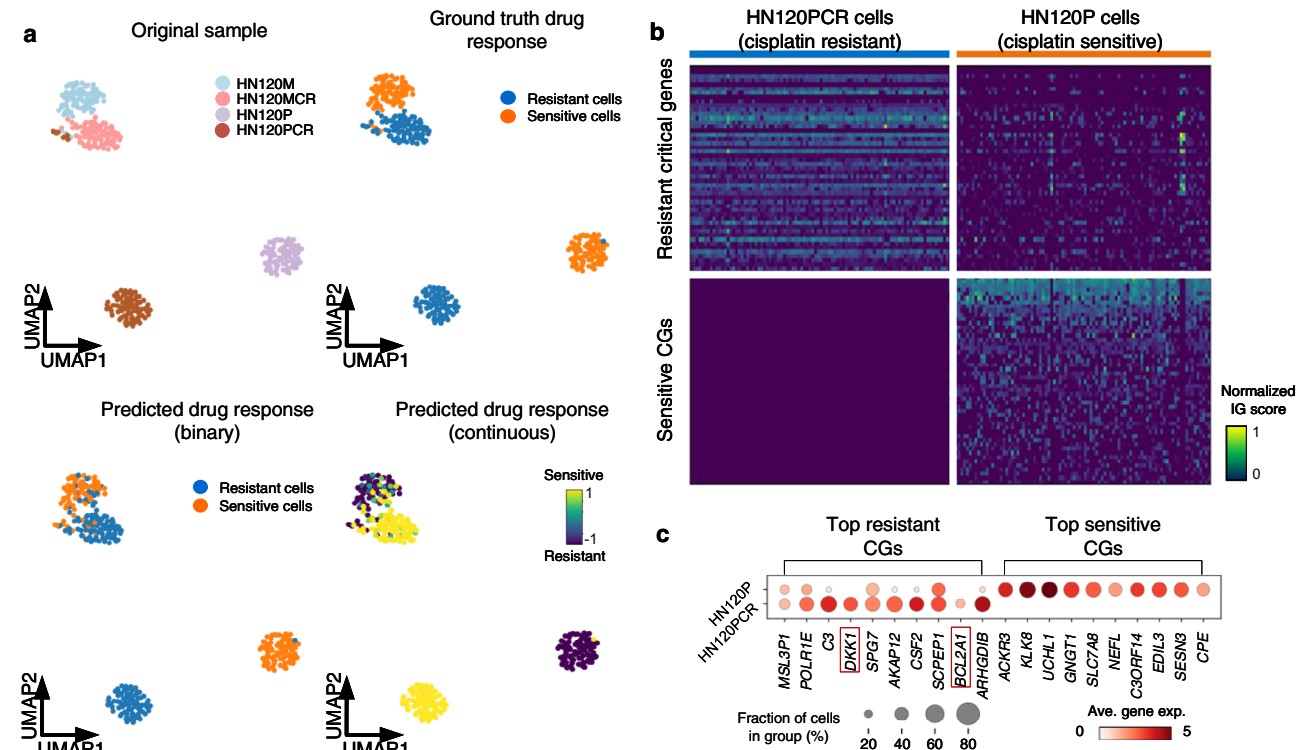

**Fig. 4 | Case study of scDEAL on Data 1 with Cisplatin drug responses. a** UMAP comparison between ground-truth labels and predicted binary response labels in scDEAL. **b** Integrated gradient heatmap of top 50 CGs in the HN120P cell group (sensitive) and HN120PCR (resistant) cell group. CGs in HN120P were considered as sensitive CGs, and CGs in HN120PCR were considered as resistant CGs. **c** Cell fraction and normalized expression levels of the top ten CGs in HN120P and HN120PCR cell groups. Abbreviation: integrated gradient score (IG), critical genes (CGs).

adjusted *p*-value = 0.003), contains multiple genes involved in cell cycle checkpoint, e.g., *CCNF*[39], *BUB1B*[40], *BUB1*[40], and *CDC25C*[41]. The activated cell cycle checkpoint can protect cells from Cisplatin-induced cell death and plays a critical role in Cisplatin resistance[42]. In addition, a few pathways that also have associations with Cisplatin resistance, such as "negative regulation of cell death"[43], "response to hypoxia"[44], and "mitotic cell cycle checkpoint"[45], are also identified from the resistant CGs, further validating that our scDEAL is able to identify genes that are important to drug responses (Supplementary Data 2).

### scDEAL drug response prediction is highly correlated with pseudotime analysis

We applied Monocle3[46] for the trajectory inference on Data 6 (treated with I-BET) to validate whether our predicted drug response is correlated with the progression of drug treatment. The gene expression-based pseudotime analysis showed a trajectory trend starting from DMSO samples towards the 1000-ml I-BET treated samples (Fig. 5a). When comparing the pseudotime results with the drug response (continuous probability score) on the same diffusion UMAP, we observed an increased resistance from DMSO control towards the treated samples (Fig. 5b). Such results indicate that the remaining living cells sequenced after high drug doses exhibited significant drug tolerance, which also aligns well with the experimental drug-response labels (ground-truth labels). In addition to the consistency between prediction and the trajectory topology, we further explained the trend of resistance development by CGs identified in scDEAL. We showcased the expression values of two representative I-BET resistant CGs, i.e., Eid2 and *Galnt17* (Fig. 5c), and two representative I-BET sensitive genes, i.e., *Emilin1* and *Ramp1* (Fig. 5d). We observed that the expression levels of these genes matched the trajectory of pseudotime analysis and predicted drug response probability scores.

Further investigation regarding the comparisons of predicted CGs and DEGs along with the trajectory indicated that the predicted CG lists have more distinct expressions in separating sensitive and resistant cell states (Fig. 5e and Source Data 9). The Pearson's correlations between scores and the pseudotime value are as high as 0.81 (positively correlated; resistant probability score vs. pseudotime) and −0.93 (negatively correlated; sensitive probability score vs. pseudotime), which indicated that predictions of scDEAL could imply drug response development. The top ten CGs in the sensitive and resistant cell group showed distinct expression patterns and were highly correlated with the pseudotime score (Fig. 5f and Source Data 10). In summary, we confirmed that the drug response results and CGs predicted in scDEAL have strong correlations to the I-BET treated cell pseudotime trajectory.

## Discussion

scDEAL augments scRNA-seq data analyses and interpretation using bulk gene expression data, which can be applied to predict drug responses of cell populations in cancer scRNA-seq data and other diseases. The neural networks adapted to scRNA-seq data can be preliminarily trained on a large volume of bulk cell-line data. Consequently, drug sensitivity can be predicted from scRNA-seq data. Noted that, scDEAL predicts single-cell drug response solely based on the trained model and scRNA-seq gene expression matrix, and no labels (neither cell type nor drug response) are needed. We performed comprehensive analyses to benchmark scDEAL on six drug-treated scRNA-seq data with experimentally validated drug-response labels. Our results indicate that scDEAL performs well and is robust in drug response label prediction and gene signature identification. We reason that AMI and ARI scores are sensitive to mislabeling, especially with only two categories in the data (sensitive and resistant); hence, they showed relatively lower scores in Data 1 and 2, and the variation of

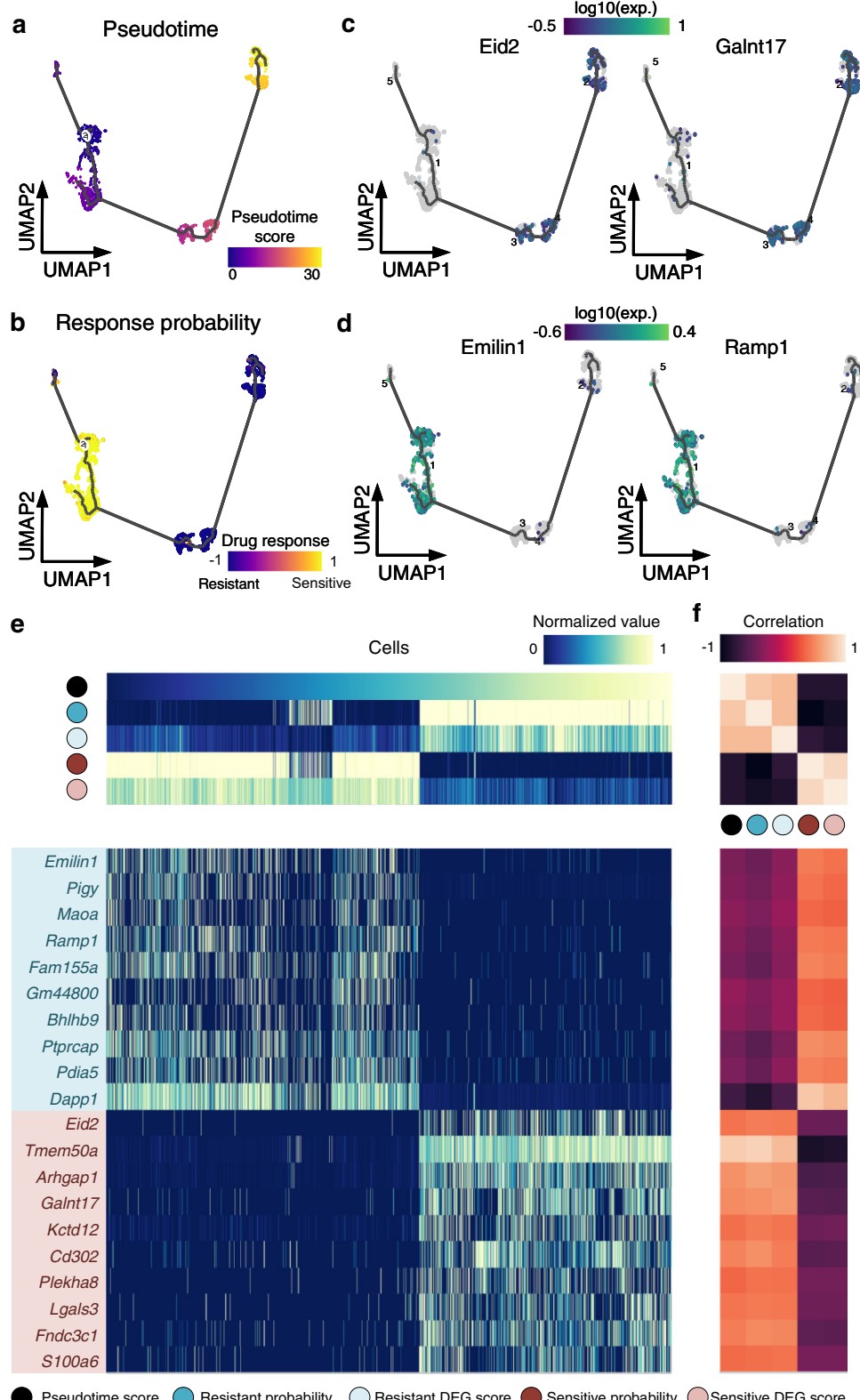

**Fig. 5 | Validating predicted drug response with pseudotime trajectory. a** Cell UMAP plot colored as per pseudotime scores predicted from the raw scRNA-seq data (Data 6). **b** Same UMAP plot colored as per predicted continuous drug response probability scores in scDEAL. **c, d** Diffusion UMAP colored by gene expressions of two representative genes in the predicted sensitive CG list and resistant CG list, respectively. **e, f** DEG scores in the sensitive and resistant cell groups, respectively, Pearson's correlations between diffusion pseudotime value, sensitive and resistant response probability. Source data are provided as Source Data 9: Pseudotime score, resistant probability, resistant score, sensitive probability, and sensitive score for each cell; Source Data 10: Pearson's correlations among pseudotime score, resistant probability, resistant score, sensitive probability, sensitive score, top 10 sensitive CGs, and the top 10 resistant CGs. Abbreviation: differentially expressed gene (DEG).

these two metrics is higher than that of the other five metrics (F1-score, AUROC, AP, precision, and recall) among six datasets. We also conducted comparative analysis to showcase the rationale of scDEAL framework design, including the contribution of DTL method, bulk data usage, autoencoder selection, and cell-type regularization. We identified CGs corresponding to the Cisplatin responses in OSCC, showing distinct predicted response patterns in drug-sensitive and -resistant cells. Our results demonstrated highly correlated drug response predictions with single-cell pseudotime analysis.

The accuracy of the prediction results in scDEAL could vary, depending on the collection of bulk gene expression of cell lines. We will update scDEAL training data by integrating additional bulk-level databases in the future. Furthermore, increased experimentally validated drug response scRNA-seq data would help determine better model hyperparameters and even help develop direct single-cell-to-single-cell deep transfer learning models. Databases, such as DrugCombDB[47], can be included to predict combinatory drug responses. In addition, the genetic features between bulk and scRNA-seq data can be explained and biologically interpreted using the IG score and CG identification. The predicted CGs can be used as targets for experimental validations on drug–gene relations via single-cell Perturb-seq[48].

Noted that a recent study pointed out a potential sample swap between Data 1 and 2 (HN120 and HN137)[49]. To test the reliability of our method, we carried out a similar drug resistance analysis by re-assembling the HN120 and HN137 datasets as suggested in the above study. Specifically, we reassembled a new HN120 data with cells originally labeled as HN137P, HN120PCR, HN120M, and HN120MCR, and a new HN137 data with cells originally labeled as HN120P, HN137PCR, HN137M, and HN137MCR. A similar grid-search method was applied to optimize the reassembled data analysis using scDEAL. The new drug response prediction F1-score of re-assembled HN120 data is 0.750 (originally 0.839) and of re-assembled HN137 data is 0.764 (originally 0.765) (Supplementary Fig. S9). The results indicated that, regardless of data re-assembly, scDEAL achieves competitive response prediction to Cisplatin, which demonstrates the reliability of our approach. Additionally, CGs identified in the re-assembled data between HN120P and HN120PCR aligned well with our previous results. All the important CGs related to Cisplatin resistance are also found in the re-assembled data. Such results indicate that scDEAL can find the CGs to drug response even though the sensitive and resistant tissues are derived from different patients and showcased the potential usage of scDEAL for combined data from different patients.

One remaining challenge in single-cell drug response prediction is the prediction across different species. Considering the genetic variation between human and mouse, drug response in one species cannot be directly transferred to predict the other. In our case study of mouse scRNA-seq data (Data 6), only homologous genes were kept in the mouse data and the scDEAL model was still trained with human cancer cell lines. Due to the limited number of drug-treated mouse benchmark scRNA-seq data in the public domain, we could not systematically evaluate and optimize the trans-species reliability in the current study. Such a topic is of significant interest in the single-cell field[50], and will be one focus of our future study along this research direction.

In summary, scDEAL has considerable potential for improving drug development at the single-cell level. First, it can be used to predict drug responses and link gene signatures with treatment effects. Second, the CGs are potential target signatures that can be used for CRISPR screening or cell reprogramming. Third, it can be applied to existing non-drug-treated scRNA-seq data to predict the potential drug response in multiple cell clusters that can be selected for animal drug tests. In the long run, we believe our work can contribute towards and provide insights into cell reprogramming, drug selection and repurposing, and combinatory drug usage for improving therapeutic efficacy.

## Methods

### Datasets

The GDSC database [https://www.cancerrxgene.org/] is publicly available online. Drug response annotation, including half maximal inhibitory concentration (IC50) and area under the dose-response curve (AUC) are available online [https://www.cancerrxgene.org/downloads/bulk_download]. Gene expression data (RMA-normalized basal expression profiles) for cell lines can be accessed on GDSC [https://www.cancerrxgene.org/gdsc1000/GDSC1000_WebResources/Home.html]. We also collected the CCLE cell line expression profile and PRISM cell line viability assay. Expression profiles and viability assays can be downloaded from [https://depmap.org/portal/]. Considering the discrepancies between the two databases, we integrated the two databases by keeping the shared genes, as well as all cell lines and drugs without missing values. Overall, we collected bulk-level drug response data with 1280 cancer cell lines, 1557 drugs/chemical compounds, and their expression profiles on 15,962 genes (Supplementary Table S4-S5).

All scRNA-seq data analyzed in this study are publicly available. The following datasets are available from the National Center for Biotechnology Information's (NCBI) Gene Expression Omnibus (GEO [https://www.ncbi.nlm.nih.gov/geo/]). All data are accessible through the GEO Series accession numbers in Data Accessibility and Supplementary Table S1. All metadata is available in the supplementary information files from the original publication.

### Preprocessing of drug response labels at the bulk level

The drug response labels of cell lines in the bulk data are derived from the response AUC values in the GDSC and CCLE databases. For each drug, we used the same method in the CCLE study[15], to binarize the AUC scores among bulk samples and determine whether the drug is sensitive or negative to the sample. Cell lines sensitive to a specifically selected drug are annotated as 1, whereas the resistant lines are annotated as 0. The waterfall method sorts cell lines according to their AUC values in descending order and generates an AUC-cell line curve in which the x-axis represents cell lines, and the y-axis represents AUC values. The cutoff of AUC values is determined via two strategies: 1) for linear curves (whose regression line fitting has a Pearson correlation >0.95), the sensitive/resistant cutoff of AUC values is the median among all cell lines; 2) otherwise, the cutoff is the AUC value of a specific boundary data point, which has the largest distance to a line linking two datapoints having the largest and smallest AUC values.

### Data sampling for predictor training

The proportion of sensitive and resistant cell lines is different across different drug treatments. This imbalance issue of drug response labels in the training set may affect the model's performance. We, therefore, introduce other sampling methods to balance the proportion of sensitive and resistant cell lines when training the prediction model at the bulk level. Three sampling methods as hyperparameters are introduced in the bulk model training, including up-sampling, down-sampling, and SMOTE-sampling[51]. Up-sampling randomly duplicates samples in the minority, and down-sampling discards samples in the majority class to generate a training set with the same number of sensitive and resistant cell lines. SMOTE generates synthetic cell lines by selecting $k$ nearest neighbors for a random sample in the minority class and synthesis of an artificial sample within neighbors in feature space. All sampling methods are implemented by the Python library imblearn[52].

### Pre-processing for scRNA-seq data

Quality control and preprocessing of the scRNA-seq data were performed using the Python package SCANPY[53]. Specifically, cells with less than 200 detected genes (indicative of no cell in the droplet), and genes detected in less than three cells were filtered out using the

function "filter_cells" and "filter_genes". Cells with a percentage of expressed mitochondrial genes higher than 10% were removed. Counts matrices were normalized by dividing by the total UMI count in each cell, multiplied by a factor of 10,000 using the function "normalize_total", and log one plus transformed using the function "log1p". Expression values are then scaled using "preprocessing.MinMaxScaler" in the package sklearn[54].

## scDEAL workflow design

The whole framework of scDEAL can be treated as two parts: supervised learning to build a model to predict the response label classification at the bulk level and using this model for label prediction at the single-cell level. We first split 64%, 16%, and 20% of bulk RNA-seq data as the training set, validation set, and testing set, respectively. These subsets were used to train the initial model in Steps 1 and 2, detailed below. The trained bulk model and the complete scRNA-seq data followed Steps 3-5 to predict drug response at the single-cell level. Since there is no ground-truth label for single-cell data in this case, there is no need to split each scRNA-seq data into training, validation, and testing subsets. The split was performed by the function "train_test_split" in the package sklearn.

**Step 1. Bulk feature extraction.** The first step of transfer learning is the gene feature extractor. Gene feature extractors are applied to extract variable gene features, reduce data dimensionality, and denoise data. Furthermore, it is a fined-tuned preliminary training step for determining the initial model weights in predictor ($P$) in step 3.

We applied a DAE to learn a low-dimensional representation from the bulk expression matrix $\mathbf{X_b}$, where each row of the matrix represents a cell line and each column of the matrix is a gene. The basic architecture of a DAE is composed of three parts:

(i) a noise operation ($B$), which generates a noisy bulk expression matrix $\mathbf{X_b}'$ by inducing random noises to $\mathbf{X_b}$ based on a binomial distribution:

$$X_b' = B(X_b, p_b), \tag{1}$$

where $p_b$ is the probability of zero value assignment in each row;

(ii) an encoder ($E_b$), which subtracts $\mathbf{X_b}'$ to a lower-dimensional subspace with a ReLU activation function; and

(iii) a decoder ($D_b$), which reconstructs an approximation matrix $\mathbf{X_b}''$ from the low-dimensional representations.

The DAE is optimized by the reconstruction loss function (Mean Square Error, MSE) between the input $\mathbf{X_b}$ and $\mathbf{X_b}''$, aiming to make the reconstructed matrix similar to $\mathbf{X_b}$. The model can be trained as:

$$\min loss_{recon}(E_b, D_b, X_b) = \min(MSE(X_b, X_b'')) \tag{2}$$

$$X_b'' = D_b(E_b(X_b')) \tag{3}$$

**Step 2. Bulk drug response prediction.** We train an encoder model (predictor, $P$) based on a fully connected multi-layer perceptron (MLP)[55] on bulk RNA-seq data to estimate the correlation of drug response and bulk gene expressions. Parameters inside $P$ are optimized with the classification loss (i.e., cross-entropy) between the predictive drug responses classification per cell line $\mathbf{Y_b}$ and the binary drug response label $\mathbf{Y_b^0}$ extracted from the bulk database.

$$\min loss_{class}\left(P, Y_b, Y_b^0\right) = \min\left(Cross\ Entropy\left(Y_b, Y_b^0\right)\right) \tag{4}$$

$$Y_b = P(E_b(X_b)) \tag{5}$$

**Step 3. Single-cell feature extraction.** A similar DAE model is also trained to extract low-dimensional features from the query scRNA-seq data $\mathbf{X_s}$. Its loss function can be defined as:

$$\min loss_{recon}(E_s, D_s, X_s) = \min(MSE(X_s, X_s'')), \tag{6}$$

$$X_s'' = D_s(E_s(X_s')), \tag{7}$$

$$X_s' = B(X_s, p_s), \tag{8}$$

where $\mathbf{X_s}'$ is the noisy single-cell expression matrix after inducing random noises with $p_s$. $E_s$ and $D_s$ are the encoder and decoder, and $\mathbf{X_s}''$ is the reconstructed scRNA-seq matrix resulting from $D_s$.

**Step 4. DTL model training.** The DTL training adapts the gene features extracted from bulk and single level to enable the sensitivity prediction for cells through the predictor $P$. We applied a DaNN[12] model to derive the feature extractor $E_s$ in the single-cell level. The DaNN model introduces an extra loss named the Maximum Mean Discrepancy (MMD) to estimate the similarity between output $E_b$ and $E_s$, which is defined as:

$$loss_{MMD}(E_b(X_b), E_s(X_s)) = |\frac{1}{n}\sum_{i=1}^{n}\phi(x_b^i) - \frac{1}{m}\sum_{j=1}^{m}\phi(x_s^j)|_H, \tag{9}$$

where $X_b = \left\{x_b^i\right\}_{i=1,...,n}$ and $X_s = \left\{x_s^j\right\}_{j=1,...,m}$ are data vectors for $n$ cell lines and $m$ cells from the bulk and scRNA-seq data, respectively; $\phi(.)$ is referred to the feature space that maps to the universal Reproducing Kernel Hilbert Space (RKHS). The RKHS norm $|.|_H$ measures the distance between two vectors with different dimensions. The similarity between two gene features is added to the classification loss during the training process of the predictor $P$ to ensure that the feature space of $E_s$ and $E_B$ have similar distributions. The DaNN model is trained to update two gene extractors ($E_B$ and $E_s$) and the predictor $P$, simultaneously, which can be defined as:

$$\min_{E_b, E_s, P} loss_{DaNN}(X_b, X_s, E_b, E_s, P) = loss_{class}(P, E_b, X_b, Y_b)$$
$$+ \alpha * loss_{MMD}(E_b(X_b), E_s(X_s)) + \beta * regulizer, \tag{10}$$

$$regulizer = \sum_{CC} \frac{cosine_{similarity}}{c\ in\ CC}(X_s), \tag{11}$$

where $\alpha$ is a weight of $loss_{MMD}(.)$, $\beta$ is the weight of the regulizer, $c$ is cell, and $CC$ is the cell cluster categories obtained from Louvain clustering results (using the igraph R package v1.3.4). By minimizing $loss_{DaNN}(.)$, the trained $E_s$ and $P$ will then be used for predicting drug responses from the scRNA-seq data.

**Step 5. Model transfer and single-cell drug response prediction.** The well-trained $E_s$ and $P$ in Step 4 will be assembled and transferred to predict the single-cell drug responses using all cells in the scRNA-seq data. The assembled $E_s$ and $P$ will take scRNA-seq data $\mathbf{X_s}$ as input and output the continuous probability scores $\mathbf{Y_s}$ for each cell in $\mathbf{X_s}$. The binary label is determined by counting any cells with a continuous probability score between 0–0.5 as resistant cells and 0.5–1 as sensitive cells.

## Benchmarking metrics for the scDEAL prediction

To evaluate the prediction of scDEAL, we applied seven metrics as shown below.

***Precision*** represents the ability of the model to correctly predict positive numbers among all positive predictions. We implemented

Precision tests using the "precision_score" function in the package sklearn.

$$Precision = \frac{True\ positive}{True\ positive + False\ positive} \quad (12)$$

**Recall** represents the model's ability to correctly predict positivity from actual positive samples. We implemented Recall tests using the "recall_score" function in the package sklearn.

$$Recall = \frac{True\ positive}{True\ positive + False\ negative} \quad (13)$$

**F1-score** can be interpreted as a weighted average of precision and recall. F1-score reaches its highest value at 1 and lowest score at 0. The equation for the F1-score is:

$$F1 - score = \frac{True\ positive}{True\ positive + 0.5*(True\ positive + False\ negative)} \quad (14)$$

We implemented F1-score tests using the "F1_score" function in the package sklearn[54].

**AUROC score** computes the area under the receiver operating characteristic (ROC) curve. The ROC curve's x-axis is the true positive rate and the y-axis is the false positive rate derived from prediction scores. The curve is generated by setting different thresholds to binarize the numerical prediction scores. AUROC computes the area under the precision-recall curve with the trapezoidal rule, which uses linear interpolation. We implemented AUROC tests using the "roc_auc_score" function in the sklearn package.

**AP score** summarizes a precision-recall curve (PRC) as the weighted mean of precisions achieved at each threshold, with the increase in recall from the previous threshold used as the weight. The AP score is given by:

$$AP = \sum_{i=1}^{n} (R_n - R_{n-1})P_n, \quad (15)$$

where $P_n$ and $R_n$ are the precision and recall at a threshold $n$ ordered by its value. We implemented AP tests using the "average_precision_score" function in the package sklearn.

**AMI** is an adjustment of the Mutual Information (MI) score to account for chance. It accounts for the fact that the MI is generally higher for two clusters with a larger number of clusters, regardless of whether there is more information shared. We implemented ARI tests using the "adjusted_mutual_info_score" function in the package sklearn.

**ARI** reaches its highest value at 1 and lowest score at 0. The equation for ARI is:

$$ARI(P^*, P) = \frac{\sum_{i,j}\binom{N_{ij}}{2} - \frac{\left[\sum_i\binom{N_i}{2}\sum_j\binom{N_j}{2}\right]}{\binom{N}{2}}}{0.5*\left[\sum_i\binom{N_i}{2} + \sum_j\binom{N_j}{2}\right] - \frac{\left[\sum_i\binom{N_i}{2}\sum_j\binom{N_j}{2}\right]}{\binom{N}{2}}}, \quad (16)$$

where $N$ is the number of points in a given data, $N_{ij}$ is the number of points of class label $C_j^* \in P^*$ assigned to cluster $C_i$ in partition $P$. $N_i$ is the number of points of cluster $C_i$ in partition $P$. $N_j$ is the number of points of cluster $C_j$. We implemented ARI tests using the "adjusted_rand_score" function in the package sklearn.

## Differentially expressed gene score
To select the differentially expressed genes, we performed Wilcoxon signed-rank tests using the "rank_genes_groups" function. By default, the top 50 genes (BH adjusted $p$-values <0.05) ranked by the z-score are selected to calculate the sensitive or resistant gene score. The sensitive or resistant gene score of each cell is calculated by subtracting the average DEG expression with the average expression of a randomly sampled reference set of genes in that cell. The raw gene score is then min-max scaled. The DEG score was evaluated with the "score_genes" function built in SCANPY[53].

## Data sampling and repetition for stability tests
The stability test is performed by retraining the DTL model ten times with different random seeds and randomly sampled subsets of cells. To preserve the original sensitive and resistant cell ratio in the dataset, we choose to perform the stratified sampling of the sensitive and resistant cells in the dataset. The sampling is performed using the "resample" function in the sklearn package[54]. The number of output (n_samples parameter) is set to be 80% of the input, and the sampled data will not be sampled again with the setting "replace = False".

## Critical gene identification with Integrated Gradients
We applied IG score[56] to characterize critical input genes features in the scDEAL model. An IG score represents the integral of gradients with respect to each gene expression as inputs along the path from zero expression as a baseline to the input expression level (Supplementary Fig. S5). The integral is approximated using the Riemann rule described as follows:

$$IG_i(x):: = (x_i - x_i') \times \int_{\alpha=0}^{1} \frac{\partial F(x' + \alpha \times (x - x'))}{\partial x_i} d \quad (17)$$

It calculated the importance of the $i$-th gene expression of the input cell $x$. $\alpha$ is the scaling coefficient; $x_i'$ is the baseline expression level gene $i$, which is 0 in our case; and $\partial F(x) / \partial x_i$ represents the gradient of $F(x)$ along the $i$-th dimension.

We apply the "IntegratedGradients" class in the Python Captum library[57] to calculate IG values. The inputs are our expression matrix, trained model, and output labels. The outputs of the function are IG matrices of the same shape as the input expression matrix. Rows represent genes and columns represent cells. Values in a matrix are the corresponding IG values.

As scDEAL is a binary classification deep learning model, it has two nodes in the output layer to predict sensitive and resistant probabilities. Based on the sensitivity or resistance for each gene contribute, we can obtain two separate IG matrices for each input data corresponding to the sensitive and resistant output. The IG matrix can be found in the model output file "attr_integrated_gradient.h5ad". The IG matrix is stored with an "AnnData" object and can be read by the function "sc.read_h5ad".

To select genes that have significantly higher IG values within the sensitive (or resistant) cell cluster, we utilized the Wilcoxon test using the function "sc.tl.rank_genes_groups" between sensitive (or resistant) cells in SCANPY. We considered the genes with Bonferroni adjusted $p$-values <0.05, log-fold changes > 0.1, and the percentage of cells with IG scores in either group higher than 0.2 as CGs.

## Functional enrichment
Functional enrichment test was performed via DAVID online service. The GOTERM_BP_DIRECT database was used for GO pathways enrichment test. Results were filtered by the default $p$-value <0.1 on DAVID.

## Trajectory inference for Oral Squamous Cell Carcinomas
The trajectory inference for the scRNA-seq data was preprocessed using Monocle3[46]. The read count matrix was projected to a 2-dimensional UMAP space using the "reduce_dimension" function. We then used the function "learn_graph" to construct a graph topology from the reduced dimension space based on the reversed graph

embedding algorithm. Afterwards, we calculated pseudotime values for cells based on their projection on the graph learned in the "learn_graph" function using the "order_cells" function.

## Reporting summary

Further information on research design is available in the Nature Research Reporting Summary linked to this article.

## Data availability

GDSC is publicly available through the website (https://www.cancerrxgene.org/). Drug response annotation, including half maximal inhibitory concentration (IC50) and area under the dose-response curve (AUC), are available through the page https://www.cancerrxgene.org/downloads/bulk_download. Gene expression data (RMA-normalized basal expression profiles) for cell lines can be accessed on GDSC (https://www.cancerrxgene.org/gdsc1000/GDSC1000_WebResources/Home.html). The CCLE cell line expression profile and PRISM cell line viability assay are available from https://depmap.org/portal/. The six scRNA-seq data used in this study are available in the GEO database without access restrictions under accession code GSE117872, GSE112274, GSE140440, GSE149383, and GSE110894. Detailed descriptions of scRNA-seq data used in this study can be found in Supplementary Table 1. Source data from each figure are provided in this paper.

## Code availability

The source code of scDEAL is freely available on (https://github.com/OSU-BMBL/scDEAL). The code is also available on Zenodo[58].

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

## Acknowledgements
This work was supported by funding from the National Institutes of Health (R01-GM131399 Q.M. and R35-GM126985 D.X.). The work was also supported by the award NSF1945971 (Q.M.) from the National Science Foundation. This work was supported by the Pelotonia Institute of Immuno-Oncology (PIIO). The content is solely the responsibility of the authors and does not necessarily represent the official views of the PIIO. The authors would like to thank Dr. Fei He (Northeast Normal University, China) for his assistance and advice in the pipeline construction, and Mr. Zhenyu Wu (Ohio State University, USA) and Ms. Ren Qi (Tianjin University) for their help in preliminary testing.

## Author contributions
Q.M. conceived the basic idea and designed the framework. J.C. and A.M. led framework construction and performed data analysis. J.C. and X.W. performed the benchmarking experiments. Q.W. and A.M. carried out the Cisplatin case study. D.X. provided valuable advice for the deep transfer learning model development and optimization, B.L. helped polish the Method section, and L.L. helped interpret drug response case studies. All the authors participated in writing the manuscript.

## Competing interests
The authors declare no competing interests.
