## [Peer review file · Nature Communications]

REVIEWER COMMENTS

Reviewer #1 (Remarks to the Author): Expert in cancer drug response and single-cell RNA-seq

Predicting therapeutic response in individual cells is challenging because the available drug sensitivity data originate from bulk RNA studies. The authors of this manuscript propose a computational framework that integrates bulk RNA data with scRNA-seq data in order to provide drug response prediction at single cell level. It also enables to connect drug response to different cell states defined by pseudotime trajectory.

Main points:

1. The study is a good starting point but lacks rigorous validation of the method. These include an in-depth analysis of discrepancies between the predicted and experimental sensitivity in all six datasets, collecting data from experiments with drug combinations as in Aissa dataset, running data on other bulk RNA data such as DrugCombDB and LINCS.
2. Performing analysis on GO and KEGG datasets is diluting the main message. There are many other programs that perform this task. Generating signatures from bulk expression profiles that work for single cells is the main challenge and this is what needs rigorous validation.

Supplementary Fig. S1, Fig. 3D and E are presented as critical figures in the paper but the results are not validated by other methods (from Sharma or others). It is unclear whether the analysis provided by the authors is any better.

p.7 Starting from the vague link between ROS and “carbon metabolism” pathways till the end of the paragraph, there is little evidence to support authors’ conclusions.

Minor points:

p.2 Provide references for “six datasets with drug treatment” as this is the first time they are mentioned

Fig. 1B. The scRNA-seq datasets were split in three parts each. Providing data for validation subsets and testing subsets will improve the clarity.

p.5 Describe the meaning of ground truth

p.6 Figure legend Fig.2B should be modified to make it more clear. "Drug response" title is confusing. Might be to change it to three panels, Ground truth, Prediction and Sensitive gene score? Describe what different colors present from left (i.e. Cell states) to right.

From p.7, what is "CGs"? It is worth making a transition from the high variable genes in Fig. 1 to CGs. It is rather an unconventional abbreviation. Also, it is unclear what criteria have been used for "resistant" and "sensitive" CGs.

p.13 Aissa the Species column should be changed to Homo sapiens

Avoid the phrase "for the first time" as well as the phrase "a broad range of bulk and scRNA-seq data", when referring to only six scRNA-seq datasets.

Rephrase:

p.5 "Though scDEAL delivered decent results on metrics comparing predictions and binarized drug response labels. The comparison cannot reflect the expression patterns of MA9 leukemic cells and my struggle in cells having sensitive probabilities in the borderline."

p.5 "the performance if scDEAL"

p.7 "an indicator of the increase of Cisplatin resistance"

p.13 "we selected 1,018 cell lines and 17,419 genes from the expression profiles correspond to a subset of drugs..., corresponding to the drug treatment"

Reviewer #2 (Remarks to the Author): Expert in single-cell RNA-seq, drug response, and cancer genomics

Summary:

Chen et al. present scDEAL, a novel deep learning-based method for transferring knowledge extracted from drug responses in bulk RNA-seq data to scRNA-seq data. scDEAL allows predicting drug responses from gene features in single cell data, without requiring annotated single cell data for training. Instead, the proposed methodology leverages the large amount of bulk RNA-seq data available following drug

treatment and a transfer learning methodology. scDEAL also includes an interpretability method based on Integrated Gradients to highlight which gene features are deemed more important by the model for classifying a cell as sensitive or resistant. The proposed method is evaluated on six scRNA-seq datasets. The analysis includes a validation of the predictions and the interpretations, and it is also supported by pseudotime analysis. This work is in an area that is timely and important, and the paper will be of interest to the broad readership of Nature Communications. While overall the study is well done and described, we do have some comments that should be addressed in a revision, to fully describe the method, address some technical questions and properly assess the limitations exposed by some of the analyses so that users are guided well when using such a method.

Major comments:

Model.

Overall, the scDEAL model explanation is clear and detailed. However, few points should be better clarified/extended:

The initial explanation of the method suggests that labels for scRNA-seq data are never used during training. However, at p. 16 l. 455, Ys (i.e., labels for scRNA-seq data) is specified as output. Is that a typo? The authors must clarify this point.

In step 4 of the methodology (transfer learning), it is described that both the encoders (Mb and Ms) and the predictor P are jointly updated. However, by doing so, the model could minimize the MMD loss simply by setting Ms equal to Mb (without modifying Mb). If this is the case, the resulting model would be equal to the model obtained after step 3. The authors must clarify this point, and in particular:

- 1) Add a paragraph explaining the rationale behind this transfer learning strategy and the expected outcome, in terms of how Mb/Ms/P are expected to be updated by this step at a high level.

- 2) Compare the results of the final model (Figure 2A) to the results obtained on the same datasets directly applying the bulk-level model (step 3). This is also crucial to define a baseline and quantify the improvement given by the transfer learning step.

The model is characterized by several hyperparameters. In addition to standard ones (learning rate, number of hidden units, etc.), the authors mention different important options for data sampling (p. 14) and the architecture of the autoencoder (p. 15). However, a discussion about these choices and their impact is missing. For example, the authors mention that both a standard autoencoder and a VAE can be used, and, looking at table S3, different choices have been made for different test datasets. We ask that the authors add a discussion about how these choices have been made, and, more importantly, about the robustness of the proposed methodology concerning these choices.

In addition, following on this point, it is important that the authors show for at least one or two dataset two alternative choices and the extent to which performance varies to help guide future users of the method and give a sense of the method's stability.

Results and analyses.

The results and the analyses are generally quite comprehensive, however, several points should be clarified and extended, especially as it is easy to create a too rosy impression:

The authors discuss and validate the predictions of the model and its interpretations through different analyses. However, the manuscript is lacking a more detailed analysis of the model's errors.

1. For example, looking at figure 3A, we see that most of the errors of the models are related to entire clusters. It would be interesting to understand what makes a certain cluster "harder" for the model. Which additional training data (bulk and/or single cell) would help the model better classifying that cluster? Such analysis would also be extremely useful as a tool to drive the collection/generation of new data. I suggest extending this part.

2. Similarly, in Figure 2D, the model has a miss in the strength of the resistant scores, compared to the ground truth and understanding its source is important.

3. In Figure 2C, we ask for caution in interpreting the scatter plots with a correlation coefficient. Given the zero inflation in scRNA-seq data along with the many genes whose expression does not change, such plots can often look better than what is meaningful. Can the authors add an appropriate null model (empirical, driven by this particular data) to get a sense?

4. Figure 2B: the text (lines 146-148) introduced four cell states for this dataset. However, figure 2B (left) includes more than 4 colors. The legend only calls out four colors, but the colors used in the plot are different with respect to those in the legend! Figure 2B (left) should be fixed or better explained, as it has left us wondering. Additionally, it is not clear what Figure 2B (right) is showing, and how it is different with respect to the central panel.

Discussion

In the Discussion, we would appreciate a bit more guidance to readers about the utility of the method for future studies.

Minor comments:

- There are few errors in words/sentences throughout the manuscript. These include: l. 99: fine-tuned -> fine-tuning, l. 123: received -> receive, l. 156: may -> my, l. 164: if -> in, l. 280: repetition of "by", l. 342: missing word at the beginning of the line. Lines 154-156: dot in the middle of the sentence. Lines 430-431: the phrase is not understandable.
- Figure 3A: the legend in the panel should include labels.
- l. 155: MA9 is not introduced.

Reviewer #3 (Remarks to the Author): Expert in deep learning and drug response prediction

The authors have applied several deep learning methods in order to "transfer" information from bulk RNA seq data to scRNA data to assist prediction (classification) tasks in the latter type of data.

Undeniably the authors have created a very useful pipeline for practitioners. However, I am missing the methodological innovation here.

The authors are mainly focusing on domain adaptation and they are using DaNN for that purpose. The MMD loss function is essentially distribution matching in the feature space and then DaNN simultaneously learns the feature extractors and prediction head. But the vanilla form of DaNN attempts to enforce all the features to be homogeneous (in the feature space, at least) such that they do not discriminate between source and target domain. But is that assumption really true?

Assuming scRNA data is offering more detailed information, forcing homogeneity may be counter-productive because information in the source domain will overwhelm the target domain if there is an unbalanced distribution. I do not see any discussion/justification on the information quality in these two types of dataset, particularly when it is well known that the noise characteristic in bulk RNA is very different from that in scRNA. As a matter of fact, the authors should at least try noisy AE's for scRNA to handle greater variation.

I am also confused by the pseudo-diffusion time graph. The authors scDEAL is built on static data. An assumption of independence is built in training phase. I do not see any dynamic structure in the methodology. how is scDEAL handling the cell-trajectory is not clear to me.

Response to Referees Letter

Reviewer #1

Predicting therapeutic response in individual cells is challenging because the available drug sensitivity data originate from bulk RNA studies. The authors of this manuscript propose a computational framework that integrates bulk RNA data with scRNA-seq data in order to provide drug response prediction at single cell level. It also enables to connect drug response to different cell states defined by pseudotime trajectory.

Main points:

(1.1). The study is a good starting point but lacks rigorous validation of the method. **(i)** These include an in-depth analysis of discrepancies between the predicted and experimental sensitivity in all six datasets, **(ii)** collecting data from experiments with drug combinations as in Aissa dataset, **(iii)** running data on other bulk RNA data such as DrugCombDB and LINCS.

Response: We appreciate your valuable comments. Here, we revised the manuscript according to your three aspects.

(i) To provide in-depth analyses of discrepancies between the predicted and experimental sensitivity in all six datasets, we first included more evaluation metrics to evaluate our predicted drug response labels in all datasets (**Supplementary Table S3**). Details of these metrics were described in Methods. Then, for each dataset, we repeated scDEAL 500 times on a set of 80% stratified sampling cells to test the robustness of scDEAL optimal parameter (**Figure 1E** and **Supplementary Figure S6**). The result shows some variations in evaluation metrics, but the variations are not large. In other words, our method is robust in different cell subpopulations. We performed a grid evaluation on six scDEAL parameters, including sampling method (upsampling, downsampling, SMOTE, and no sampling), learning rate (0.01, 0.1, 0.5), dropout rate (0.1, 0.3), bottleneck (32, 64, 128, 256, 512), encoder dimensions ([512, 256], [256, 128]), and predictor encoder dimensions ([256, 128], [128, 64]). In all, 480 parameter combinations were tested (**Supplementary Figure S5**). Last, we provided more visualizations, including UMAPs using scDEAL transfer embeddings Sankey plots (**Supplementary Figure S3**). This allows the readers to observe and compare the prediction results of scDEAL more intuitively.

(ii) We agree with the reviewer that the prediction of combinatory drug responses is critical in practices. However, the current design of scDEAL is not optimized for predicting combinatory drug responses because (a) the drug combination in the Aissa dataset (Osimertinib + Crizotinib) is not included in the DrugCombDB, and thus no suitable bulk

data can be used for the transfer training in scDEAL; (b) the benchmark drug response labels are not provided at the cell level (only shown in the original manuscript), and thus, our predictions can be hardly evaluated; (c) the limited single-cell drug combinatory studies can hardly support the optimization and robustness of scDEAL predictions. All these three situations cannot support the trustworthiness of an AI framework.

Nevertheless, we still try to see the performance of scDEAL in predicting combinatory drug responses in the Aissa dataset. Only untreated cells and cells treated with combined drugs were used, and we considered untreated cells as all sensitive and treated cells are all resistant in this experiment. For the bulk training data, we selected the corresponding expression profiles and labels of Osimertinib and Crizotinib in the GDSC and CCLE databases. We then considered cell lines resistant or sensitive to both drugs simultaneously and discarded the rest. The F1-score of our predicted response labels is 0.56 and 0.68, respectively (**Figure below**). Although the result can demonstrate our perspectives to some extent, we prefer not to mention these results in our manuscript, considering that using individual drug response to predict combinatory drug response is debatable. Also, the robustness and parameter optimization cannot be achieved based on a single dataset. Once more suitable datasets are available, we will further extend scDEAL to combinatory drugs prediction.

Figure for Comment 1.1. Comparisons of combinatory drug response predictions. (Top) Cells labeled in three states: combined resistance, single drug resistance, and drug sensitive. Our predicted result has an F1 score of 0.56, compared with the ground truth label. (Bottom) Cells labeled in two states: combined resistance and drug sensitive. Our

predicted result has an F1 score of 0.68, compared with the ground truth label.

(iii) We now have included both GDSC and CCLE bulk databases in the training of scDEAL model. Considering the discrepancies of the two databases, we integrated the two databases by keeping the shared genes, as well as all cell lines and drugs without missing values. Overall, we collected bulk level drug response data with 1,280 cancer cell lines, 1,557 drugs/chemical compounds, and the expression profiles of 15,962 genes (**Supplementary Table S1-S2**). We provided **Figure 2B** to show the substantial increase in using the combined GDSC and CCLE databases. Unfortunately, our study could not include DrugCombDB and LINCS (the two databases as suggested). DrugCombDB is a database integrating drug combinations from various data sources. As explained above, the current version of scDEAL is not designed for combinatory drug response prediction, and thus, DrugCombDB is not considered. In the LINCS database, we could not find drug response scores or labels corresponding to the expression data, and thus cannot be used for scDEAL training. We will keep an eye on these two databases, as well as others for our future work, and will integrate them when the appropriate information is available.

(1.2) Performing analysis on GO and KEGG datasets is diluting the main message. There are many other programs that perform this task. Generating signatures from bulk expression profiles that work for single cells is the main challenge and this is what needs rigorous validation.

Response: We have revised the manuscript to shorten the descriptions of this part and put efforts in validating signature genes and TFs identified in scDEAL based on literature. We also agree that identifying signatures from the bulk data that can be used for single cells is a challenging task. To demonstrate the successful transfer of signatures from bulk expression files to single cells, we provided two evaluations:

(i) We calculated an integrated gradients (**IG**) score to reflect the potential contribution of each gene to the final drug response label prediction (**Supplementary Figure S4 and Methods**). The IG score is based on the accumulation of gradients of neurons in a neural network following the path of layer connections. A gene with a higher IG score to the drug sensitive labeling indicates the gene is more related to drug sensitivity and contributes more for categorizing samples to drug sensitive labels. Similar rules were applied to the resistant labels. By comparing the number of genes contributing to the drug response labeling at both bulk and single-cell levels, we found that, on average, 46% of genes were shown overlapped in both data types contributing to drug sensitivity, and 53% genes were shown overlapped to drug resistance (**Supplementary Table S10**).

(ii) We compared the model performance of scDEAL to a model trained only on the bulk

RNA-seq data and directly used for scRNA-seq data prediction, without step 4 (transfer learning). The results performed on all six datasets showed a significant increase of F1 scores when using the transfer strategy than without it (**Figure 2B**). On average, scDEAL achieved a 35% increase in terms of F1 score compared to the model without transfer learning (**Supplementary Table S5**). Our comparison showed that the transfer learning indeed contributed and improved to the performance in single-cell drug response prediction.

(1.3) Supplementary Fig. S1, Fig. 3D and E are presented as critical figures in the paper but the results are not validated by other methods (from Sharma or others). It is unclear whether the analysis provided by the authors is any better.

Response: The original paper has a special focus on the epigenetic impacts on the Cisplatin resistance. Specifically, they concluded that the inhibition of BRD4 TF can reverse the cisplatin-resistance state, indicating that BRD4 is a sensitive marker to Cisplatin response. They also claimed that SOX2 and SOX9 are critical TFs acting as mediators to induce Cisplatin resistance. The knockdown of SOX2 and SOX9 can increase cellular sensitivity to cisplatin. In our results, we found BRD4 was ranked as the top one TF enriched to sensitive GCs, and SOX2 and SOX9 TFs were enriched in the 126th and 56th places in our sensitive TF list (Supplementary Data S2). On the other hand, CGs were not mentioned in the original paper but have been reported as either Cisplatin-resistant or -sensitive signatures. For example, UCHL1, PEG10, and MGMT were top 10 Cisplatin resistant CGs. They have been previously reported as key genes whose gene expressions are highly correlated with Cisplatin resistance (see the first three references below, which are also cited in the main text); FGF2 was the top one Cisplatin sensitive CG in scDEAL, and it has also been reported (see the fourth reference below, which are also cited in the main text).

- Jin, C. et al. UCHL1 Is a Putative Tumor Suppressor in Ovarian Cancer Cells and Contributes to Cisplatin Resistance. *J Cancer* 4, 662-670, doi:10.7150/jca.6641 (2013)
- Li, J., Wood, W. H., Becker, K. G., Weeraratna, A. T. & Morin, P. J. Gene expression response to cisplatin treatment in drug-sensitive and drug-resistant ovarian cancer cells. *Oncogene* 26, 2860-2872, doi:10.1038/sj.onc.1210086 (2007)
- Chen, S.-H. et al. O6-methylguanine-DNA methyltransferase modulates cisplatin-induced DNA double-strand breaks by targeting the homologous recombination pathway in nasopharyngeal carcinoma. *Journal of Biomedical Science* 28, 2, doi:10.1186/s12929-020-00699-y (2021)
- Makondi, P. T., Chu, C.-M., Wei, P.-L. & Chang, Y.-J. Prediction of novel target genes and pathways involved in irinotecan-resistant colorectal cancer. *PLOS ONE* 12,

(1.4) p.7 Starting from the vague link between ROS and “carbon metabolism” pathways till the end of the paragraph, there is little evidence to support authors’ conclusions.

Response: We have carefully revised the description of the pathway enrichment analysis along with some updates in **Figures 3D** and **3E**.

“Pathway enrichment analysis on predicted Cisplatin sensitive CGs revealed that these genes are involved in carbon metabolism, biosynthesis of cofactors and amino acids, glutathione metabolism, and DNA replication (Figure 3D). It has been reported that oxidative stress induced by Cisplatin can cause acute changes in carbon flux from pyruvate into lactate²⁸, which can explain the enrichment of the carbon metabolism pathway to be a Cisplatin-sensitive target. By contrast, the Cisplatin-resistant CGs were enriched in the Herpes simplex virus (HSV) infection. HSV has been reported to maintain the ability to induce cisplatin resistance in infected cells (Figure 3E).”

Minor points:

(1.5) p.2 Provide references for “six datasets with drug treatment” as this is the first time they are mentioned Fig. 1B. The scRNA-seq datasets were split in three parts each. Providing data for validation subsets and testing subsets will improve the clarity.

Response: We have added references for datasets when they are first mentioned in the main text (also attach them below for reviewer’s convenience). In addition, we apologize for the inaccurate description regarding the scRNA-seq data split. The whole framework of scDEAL can be treated as two parts: a supervised learning to build a model to predict the response label classification at the bulk level and using this model for label prediction at the single-cell level (**Figure 1**). We first train the bulk model in steps 1 and 2 using the training, validation, and testing subsets. The bulk testing subset data was provided in **Supplementary Table S4**. Then, we applied this model to predict response labels for all the cells in a scRNA-seq dataset. Since there is no ground-truth label for single-cell data in this case, there is no need to split each scRNA-seq data into training, validation, and testing subsets. On the other hand, the homogeneity of cell-line bulk data can reflect gene expression profiles of individual cell types, the training model is transferable between bulk data and scRNA-seq data.

- Sharma, A. et al. Longitudinal single-cell RNA sequencing of patient-derived primary cells reveals drug-induced infidelity in stem cell hierarchy. Nature communications 9, 4931 (2018)
- Kong, S. L. et al. Concurrent Single-Cell RNA and Targeted DNA Sequencing on an Automated Platform for Comeasurement of Genomic and Transcriptomic Signatures.

Clinical chemistry 65, 272-281 (2019)

- Schnepf, P. M. et al. Single-Cell Transcriptomics Analysis Identifies Nuclear Protein 1 as a Regulator of Docetaxel Resistance in Prostate Cancer Cells. *Molecular Cancer Research* 18, 1290-1301 (2020)
- Aissa, A. F. et al. Single-cell transcriptional changes associated with drug tolerance and response to combination therapies in cancer. *Nat Commun* 12, 1628, doi:10.1038/s41467-021-21884-z (2021).PMC7955121
- Bell, C. C. et al. Targeting enhancer switching overcomes non-genetic drug resistance in acute myeloid leukaemia. *Nature communications* 10, 1-15 (2019)

(1.6) p.5 Describe the meaning of ground truth

Response: We have added the description in the manuscript as follows:

“A ground truth label is a binary indicator (0 for resistant and 1 for sensitive) extracted from the original manuscripts. Most studies determine the drug response to a whole cell group based on treatment conditions, e.g., dimethyl sulfoxide (DMSO) treated cells are all sensitive, and cells survived after treatment are all resistant.”

(1.7) p.6 Figure legend Fig.2B should be modified to make it clearer. “Drug response” title is confusing. Might be to change it to three panels, Ground truth, Prediction and Sensitive gene score? Describe what different colors present from left (i.e. Cell states) to right.

Response: Thanks for your advice. We have revised the original **Figure 2B** (now is **Figure 3A**) and legend accordingly. Noted that, scDEAL can generate both continuous and binary predictions for drug responses. Thus, we kept four panels in **Figure 3A**. All legends have been clearly indicated in the figure.

(1.8) From p.7, what is “CGs”? It is worth making a transition from the high variable genes in Fig. 1 to CGs. It is rather an unconventional abbreviation. Also, it is unclear what criteria have been used for “resistant” and “sensitive” CGs

Response: “CG” is the abbreviation of critical genes. We first defined an integrated gradient (IG) score by tracing and accumulating integrated gradients of each neuron in the neural network. We mentioned the criteria of IG and CG in the method section:

“IG represents the integral of gradients with respect to each gene expression as inputs along the path from zero expression as a baseline to the input expression level (Supplementary Fig. S6). The integral is approximated using the Riemann rule described as follows:

$$IG_i(x) ::= (x_i - x'_i) \times \int_{\alpha=0}^1 \frac{\partial F(x' + \alpha \times (x - x'))}{\partial x_i} d\alpha \quad \text{Eq. 17}$$

It calculated the importance of the i -th gene expression of the input cell x . α is the scaling

coefficient; x_i^0 is the baseline expression level gene i , which is 0 in our case; and $\partial F(x) / \partial x_i$ represents the gradient of $F(x)$ along the i -th dimension.”

“To select genes that have significantly higher IG values within the sensitive (or resistant) cell cluster, we utilized the Wilcoxon test using the function "sc.tl.rank_genes_groups" between sensitive (or resistant) cells in SCANPY. We considered the genes with Bonferroni adjusted p-values less than 0.05 and log-fold changes higher than 1 as CGs. Sensitive CGs and resistant CGs are further determined depending on the predicted sensitive and resistant cluster labels.”

(1.9) p.13 Aissa the Species column should be changed to Homo sapiens

Response: We have made the change in **Supplementary Table S1**.

(1.10) Avoid the phrase “for the first time” as well as the phrase “a broad range of bulk and scRNA-seq data”, when referring to only six scRNA-seq datasets.

Response: Thanks for the suggestion. We have removed such descriptions.

(1.11) Rephrase

- p.5 “Though scDEAL delivered decent results on metrics comparing predictions and binarized drug response labels. The comparison cannot reflect the expression patterns of MA9 leukemic cells and my struggle in cells having sensitive probabilities in the borderline.”
- p.5 “the performance if scDEAL”
- p.7 “an indicator of the increase of Cisplatin resistance”
- p.13 “we selected 1,018 cell lines and 17,419 genes from the expression profiles correspond to a subset of drugs..., corresponding to the drug treatment”

Response: Thanks for your suggestions. We have thoroughly revised the manuscript.

Reviewer #2

Summary:

Chen et al. present scDEAL, a novel deep learning-based method for transferring knowledge extracted from drug responses in bulk RNA-seq data to scRNA-seq data. scDEAL allows predicting drug responses from gene features in single cell data, without requiring annotated single cell data for training. Instead, the proposed methodology leverages the large amount of bulk RNA-seq data available following drug treatment and a transfer learning methodology. scDEAL also includes an interpretability method based on Integrated Gradients to highlight which gene features are deemed more important by the model for classifying a cell as sensitive or resistant. The proposed method is evaluated on six scRNA-seq datasets. The analysis includes a validation of the

predictions and the interpretations, and it is also supported by pseudotime analysis. This work is in an area that is timely and important, and the paper will be of interest to the broad readership of Nature Communications. While overall the study is well done and described, we do have some comments that should be addressed in a revision, to fully describe the method, address some technical questions and properly assess the limitations exposed by some of the analyses so that users are guided well when using such a method.

Major comments:

Model. Overall, the scDEAL model explanation is clear and detailed. However, few points should be better clarified/extended:

(2.1) The initial explanation of the method suggests that labels for scRNA-seq data are never used during training. However, at p. 16 l. 455, Y_s (i.e., labels for scRNA-seq data) is specified as output. Is that a typo? The authors must clarify this point.

Response: Thanks for your comments. In the original scDEAL framework, we claimed that cell type labels (i.e., cell clustering results or meta information of cells) were never used in model training. Y_s indicates the output of drug response labels rather than cell type labels. In the revision, we included cell type information in the loss function as a regularizer. Such modification is for the response to the third comment of Reviewer 3, where he/she was concerned that the heterogeneity of scRNA-seq data might be lost without any specific processes. The new loss function is

$$\min_{E_b, E_s, P} \text{loss}_{DaNN}(X_b, X_s, E_b, E_s, P) = \text{loss}_{class}(P, E_b, X_b, Y_b) + \alpha * \text{loss}_{MMD}(E_b(X_b), E_s(X_s)) + \beta * \text{regularizer}, \#Eq. 10$$

$$\text{regularizer} = \sum_{c \text{ in } CC} \text{cosine}_{similarity}(X_s), \#Eq. 11$$

where α is a weight of loss_{MMD} , β is the weight of the *regularizer*, c is cell, and CC is the cell cluster categories obtained from Louvain clustering results.

(2.2) In step 4 of the methodology (transfer learning), it is described that both the encoders (M_b and M_s) and the predictor P are jointly updated. However, by doing so, the model could minimize the MMD loss simply by setting M_s equal to M_b (without modifying M_b). If this is the case, the resulting model would be equal to the model obtained after step 3. The authors must clarify this point, and in particular:

- 1) Add a paragraph explaining the rationale behind this transfer learning strategy and the expected outcome, in terms of how $M_b/M_s/P$ are expected to be updated by this step at a high level.
- 2) Compare the results of the final model (Figure 2A) to the results obtained on the same datasets directly applying the bulk-level model (step 3). This is also crucial to define a

baseline and quantify the improvement given by the transfer learning step.

Response: We appreciated the reviewer's valuable advice. Deep transfer learning is used to transfer knowledge and relation patterns from one model to another, to avoid the challenge of limited training data. Using transfer learning, we can solve a particular task using full or part of an already pre-trained model on a different task. Only limited drug response studies have been performed on the single-cell level and provided experimental response labels, thus not providing enough power to directly train deep learning models on scRNA-seq data. The expected main outcomes of scDEAL include a binary drug response label predicted for each cell and a group of critical genes response to either sensitivity or resistance in each cell cluster. We expect Mb/Ms/P (now are $E_b/E_s/P$) to be updated to harmonize bulk expression data and scRNA-seq data and transfer the trustworthy gene-drug relations from bulk-level to the single-cell level.

To answer the second question, we performed the comparison on all six single-cell datasets using scDEAL with or without transfer learning, as suggested by the reviewer, and showed the result in **Figure 2B**. On average, scDEAL achieved 35% increase in terms of F1 score compared to the model without transfer learning (**Supplementary Table S5**). Our comparison showed that the transfer learning indeed contributed and improved to the performance in single-cell drug response prediction.

(2.3) The model is characterized by several hyperparameters. In addition to standard ones (learning rate, number of hidden units, etc.), the authors mention different important options for data sampling (p.14) and the architecture of the autoencoder (p. 15). However, a discussion about these choices and their impact is missing. For example, the authors mention that both a standard autoencoder and a VAE can be used, and, looking at table S3, different choices have been made for different test datasets. We ask that the authors add a discussion about how these choices have been made, and, more importantly, about the robustness of the proposed methodology concerning these choices.

Response: we performed several tests to answer the reviewer's questions. First, we took the advice of reviewer 3 to use denoising AE instead of AE and VAE in bulk and single-cell data feature extraction, and the performances were enhanced (**Figure 2D**). Second, we performed a grid evaluation on six scDEAL parameters, including sampling method (upsampling, downsampling, SMOTE, and no sampling), learning rate (0.01, 0.1, 0.5), dropout rate (0.1, 0.3), bottleneck (32, 64, 128, 256, 512), encoder dimensions ([512, 256], [256, 128]), and predictor encoder dimensions ([256, 128], [128, 64]). In all, 480 parameter combinations were tested (**Supplementary Figure S5**). Our descriptions of the results are as follows:

“Overall, our results showed no significant effects of single parameter selections on scDEAL performances. Four datasets (Data 1, 2, 4, and 5) are more robust on all

parameter combinations than Data 3 and 6. The performance and robustness of scDEAL are likely to be related to the combination of parameters, but not in a sensitive fashion.”

(2.4) In addition, following on this point, it is important that the authors show for at least one or two dataset two alternative choices and the extent to which performance varies to help guide future users of the method and give a sense of the method’s stability.

Response: Based on the results in **Supplementary Figure S5** and our optimized parameters in each dataset (**Supplementary Table S2**), we recommend, for any new dataset, adjusting the bulk sampling methods and bottleneck dimensions. We found these two parameters differed a lot among six datasets when achieving the best prediction performance.

Results and analyses. The results and the analyses are generally quite comprehensive, however, several points should be clarified and extended, especially as it is easy to create a too rosy impression:

(2.5) The authors discuss and validate the predictions of the model and its interpretations through different analyses. However, the manuscript is lacking a more detailed analysis of the model’s errors. For example, looking at figure 3A, we see that most of the errors of the models are related to entire clusters. It would be interesting to understand what makes a certain cluster “harder” for the model.

Response: We have revised our scDEAL framework in the revision and the prediction result have been enhanced as shown in **Supplementary Figure S3**. We saw that the errors of the mismatching of response labels in an entire cluster only exist in Data 2 (Cisplatin-HN137). We have carefully explored the possibility that may cause such a situation, including the model design and original data labeling. It seems this data is a special case compared to the other five datasets; hence, we will keep exploring new strategies to make scDEAL more robust for various single-cell RNA-seq data.

(2.6) Which additional training data (bulk and/or single cell) would help the model better classifying that cluster? Such analysis would also be extremely useful as a tool to drive the collection/generation of new data. I suggest extending this part.

Response: We thank the reviewer for the comment. Indeed, the source of training data could help enhance the performance of a deep learning model. In our case, bulk expression data is used to train the relations between gene expressions and drug responses. For each drug, more bulk level data would be useful to help the model better classify drug response labels in the single-cell data. We have proved this point by comparing results between using only GDSC database, only CCLE database, and the combination of GDSC and CCLE databases. The results showed that in all benchmark data, the combined bulk database achieved the best prediction performances (**Figure**

2C). We also discussed more ideal training data in the Discussion section:

“The accuracy of the prediction results in scDEAL could vary, depending on the collection of bulk gene expression of cell lines. We will update scDEAL training data by integrating more bulk level databases in the future. Furthermore, more experimentally validated drug response scRNA-seq data would help determine better model hyperparameters and even help develop direct single-cell to single-cell deep transfer learning models.”

(2.7) Similarly, in Figure 2D, the model has a miss in the strength of the resistant scores, compared to the ground truth and understanding its source is important.

Response: This issue no longer exists based on our current model by replacing autoencoder with denoising autoencoder and adding a cell type regularization in the transfer learning loss function. The original **Figure 2D** has been updated as the new **Figure 3B**.

(2.8) In Figure 2C, we ask for caution in interpreting the scatter plots with a correlation coefficient. Given the zero inflation in scRNA-seq data along with the many genes whose expression does not change, such plots can often look better than what is meaningful. Can the authors add an appropriate null model (empirical, driven by this particular data) to get a sense?

Response: We appreciate the reviewer’s advice. As suggested, we provided a null hypothesis test by randomly selected gene sets containing the same number of genes as our identified differentially expressed genes in sensitive and resistant cell groups. We also calculated the correlation with gene scores based on benchmark drug response groups. The random selection was repeated 1,000 times, and for each time, we calculated the same correlation coefficient. We reported the null model results as the new **Figure 3D**. Our empirical test results showed that p-values for both sensitive and resistant DEG score correlations are lower than 0.001, indicating our correlation is statistically significant.

(2.9) Figure 2B: the text (lines 146-148) introduced four cell states for this dataset. However, figure 2B (left) includes more than 4 colors. The legend only calls out four colors, but the colors used in the plot are different with respect to those in the legend! Figure 2B (left) should be fixed or better explained, as it has left us wondering. Additionally, it is not clear what Figure 2B (right) is showing, and how it is different with respect to the central panel.

Response: We have regenerated **Figure 2B** (now is **Figure 3A**). The previous color mixture in the cell type panel was due to typos in the metadata file. Four treatment groups are included in our analysis. The ground truth drug response labels were defined

according to treatments. The right two panels reflect the binary and continuous prediction scores of drug response.

Discussion

(2.10) In the Discussion, we would appreciate a bit more guidance to readers about the utility of the method for future studies.

Response: We thank the reviewer's comment. The outlook of scDEAL has been added to the discussion section and described as follows:

“scDEAL has considerable potential in improving drug development at the single-cell level. First, it can be used to predict drug responses and link gene signatures with treatment effects. Second, the CGs are potential target signatures that can be used for CRISPR screening or cell reprogramming. Third, it can be applied to existing non-drug-treated scRNA-seq data to predict the potential drug response in multiple cell clusters that can be selected for animal drug tests. In the long run, we believe our work can contribute and provide insights for cell reprogramming, drug selection and repurposing, and combinatory drug usage for improving therapeutic efficacy.”

Minor comments:

(2.11) There are few errors in words/sentences throughout the manuscript. These include: l. 99: fine-tuned -> fine-tuning, l. 123: received -> receive, l. 156: may -> my, l. 164: if -> in, l. 280: repetition of “by”, l. 342: missing word at the beginning of the line. Lines 154-156: dot in the middle of the sentence. Lines 430-431: the phrase is not understandable.

Response: We have made corrections and revised the manuscript carefully.

(2.12) Figure 3A: the legend in the panel should include labels.

Response: We have revised the original **Figure 3A** (now is **Figure 4A**) to include color labels.

(2.13) 1. 155: MA9 is not introduced.

Response: Mixed Lineage Leukemia-AF9 (MA9) is a leukemia fusion gene formed upon translocation of the AF9 gene on chromosome 9 and the MLL gene on chromosome 11. We have added the full name in the revised manuscript.

Reviewer #3:

Expert in deep learning and drug response prediction. The authors have applied several deep learning methods in order to "transfer" information from bulk RNA seq data to scRNA data to assist prediction (classification) tasks in the latter type of data.

(3.1) Undeniably the authors have created a very useful pipeline for practitioners.

However, I am missing the methodological innovation here.

Response: we highlighted our methodological innovation in the Introduction section as follows:

“The novelty of scDEAL is as follows: (i) it takes advantage of the large amount of bulk-level drug response RNA-seq information from the Genomics of Drug Sensitivity in Cancer (GDSC) database^{13,14} and Cancer Cell Line Encyclopedia (CCLE)¹⁵ to train and optimize the model^{16,17}; (ii) to fill the gap of data structure differences between bulk and scRNA-seq data, scDEAL harmonizes single-cell and bulk embeddings to ensure the drug response labels transferable from bulk to single cells; (iii) to avoid losing heterogeneity in scRNA-seq data, scDEAL includes cell cluster labels for loss function regularization in each training epoch; (iv) the integrated gradient interpretation infers signature genes of drug response predictions, which makes our model more interpretable.”

(3.2) The authors are mainly focusing on domain adaptation and they are using DaNN for that purpose. The MMD loss function is essentially distribution matching in the feature space and then DaNN simultaneously learns the feature extractors and prediction head. But the vanilla form of DaNN attempts to enforce all the features to be homogeneous (in the feature space, at least) such that they do not discriminate between source and target domain. But is that assumption really true?

Response: The DaNN model balances the training loss between $loss_{class}$ and $loss_{MMD}$ by a hyper-parameter α , instead of enforcing features from bulk RNA-seq and scRNA-seq to be the same. We expect the framework to be updated to harmonize bulk expression data and scRNA-seq data and transfer the trustworthy gene-drug relations from bulk-level to the single-cell level. To maintain the heterogeneity on the single-cell side, we made some modifications to the model, described as follows:

“One of the critical issues in the model training is to maintain single-cell heterogeneity when harmonizing scRNA-seq data with bulk data. Two strategies were applied. First, as the noise characteristics in bulk RNA-seq and scRNA-seq data are quite different, we used a DAE model, rather than a common autoencoder or a variational autoencoder, to induce noises in bulk as well as scRNA-seq prior to the feature selection. In this manner, we could avoid the risk of imbalanced training that would only force gene expressions in scRNA-seq data close to bulk RNA-seq data. Second, we integrated cell clustering results to regularize the overall loss function of scDEAL, so that the cellular heterogeneity would be retained during the training process.”

“Third, we validated whether using DAE and cell type regularization can help reduce the

loss of single cell heterogeneity and enhance the prediction performance. We compared the results of the framework using common autoencoders for the feature extraction in bulk and scRNA-seq data, a framework using DAE but not regularized by cell type, and the final scDEAL framework (include both DAE and cell type regularization). For all six datasets, using DAE and cell type regularization in the framework achieved a better performance than the other two options (**Figure 2D** and **Supplementary Table S9**).”

(3.3) Assuming scRNA data is offering more detailed information, forcing homogeneity may be counter-productive because information in the source domain will overwhelm the target domain if there is an unbalanced distribution. I do not see any discussion/justification on the information quality in these two types of dataset, particularly when it is well known that the noise characteristic in bulk RNA is very different from that in scRNA. As a matter of fact, the authors should at least try noisy AE's for scRNA to handle greater variation.

Response: Thank you for the valuable suggestion. To answer your questions, we have revised the scDEAL framework:

(i) Modifying our DaNN loss function with scRNA-seq cell cluster information included:

$$\min_{E_b, E_s, P} \text{loss}_{DaNN}(X_b, X_s, E_b, E_s, P) = \text{loss}_{class}(P, E_b, X_b, Y_b) + \alpha * \text{loss}_{MMD}(E_b(X_b), E_s(X_s)) + \beta * \text{regularizer}, \#Eq. 10$$

$$\text{regularizer} = \sum_{c \text{ in } CC} \text{cosine}_{similarity}(X_s), \#Eq. 11$$

where α is a weight of loss_{MMD} , β is the weight of the *regularizer*, c is cell, and CC is the cell cluster categories obtained from Louvain clustering results. Cells in the same cell cluster can be considered as bulk data that are meant to maintain similar gene expression patterns, while the heterogeneity of scRNA-seq data will be preserved by the separation of different cell clusters.

(ii) Using denoising autoencoder (DAE) to replace AE and VAE in scRNA data's feature extraction. Details can be found in the answer to your last question regarding the comparison of AE and DAE. In conclusion, adding single-cell heterogeneous information and using DAE feature extractor instead of AE/VAE can improve the performance of the drug response prediction (**Figure 2D**).

(3.4) I am also confused by the pseudo-diffusion time graph. The authors scDEAL is built on static data. An assumption of independence is built-in training phase. I do not see any dynamic structure in the methodology. how is scDEAL handling the cell-trajectory is not clear to me.

Response: It is true that scDEAL is built on static data and each scRNA-seq data is

trained independently. Hence, it is called pseudo-diffusion time instead of real diffusion time. We predicted cell trajectory by applying an existing tool, Monocle3, directly on the original scRNA-seq data. The pseudo time reflects the aging of cells in the static data. In short, we used trajectory results to extend the validation and biological interpretation of drug response prediction in scDEAL. We revised the corresponding content to make it clear.

REVIEWER COMMENTS

Reviewer #1 (Remarks to the Author):

The manuscript still does not include rigorous validation of the method. The authors now better explained their method but then I am not sure I see much methodological innovation.

The uniqueness of single-cell data is in providing information about cell population heterogeneity and predicting cell clusters/subpopulations. The authors show sample-specific response (treated/untreated) but fail to demonstrate that scDEAL predicts cluster-specific response. Reviewer's 2 Point 2.6 is poorly addressed.

Reviewer's 1 Point 1.4 is not completely addressed. One example is "the carbon metabolism pathway to be a Cisplatin-sensitive target". I am not sure that I am aware of this pathway. Shouldn't it be "one-carbon"? Overall, the mentioned critical genes are quite superficially connected to cisplatin sensitivity and can't be considered as a validation of the approach.

Not clear what "a drug-sensitive marker to Cisplatin response" really means, and why BRD4 belongs to this category. There is no supporting evidence in the cited literature:

Yosinski, J., Clune, J., Bengio, Y. & Lipson, H. How transferable are features in deep neural networks? arXiv preprint arXiv:1411.1792 (2014)

Similar, what are the "mediators to induce Cisplatin resistance"?

The validated factors are positioned at the 126th and 56th places. Are the top TFs all false positive? Why don't they appear?

Data on the OSCC model is supposed to be presented in Figures 3a, 3B and 3C but lacking.

It is unclear why scDEAL drug response prediction should correlate with pseudotime analysis. Also, there is no explanation of Data 6 samples in order to assess robustness of the experimental design.

Reviewer #2 (Remarks to the Author):

We thank the authors for their thorough and thoughtful revision. Overall, the revised manuscript has addressed all the comments we previously raised. In particular, we are pleased that the authors significantly extended and improved the method description (with few key updates on the method itself), the analyses, and the presentation of the results.

We ask that a few minor points be addressed before the manuscript is accepted for publication:

- In Figure 2, the authors have added a robustness test for the performance of the model across different scRNA-seq datasets. However, the authors should also add confidence intervals (or, at least, show replicates) for all the other bar plots in Figure 2, as this would further demonstrate the robustness of the obtained results.

- The addition of the cell cluster loss is a valuable improvement compared to the previous version of the method. The authors already showed that it improves performance across datasets compared to a baseline (Fig. 2D). We ask that the authors also show how it can better preserve the heterogeneity of scRNA-seq data, as the authors claim multiple times in the manuscript. For example, the authors could show latent representations obtained with/without this additional loss.

Reviewer #3 (Remarks to the Author):

I thank the authors for providing satisfactory answers to the questions I posed.

Reviewer #1

1. The manuscript still does not include rigorous validation of the method. The authors now better explained their method but then I am not sure I see much methodological innovation.

Response: Thanks for your comments. We have now included additional experiments and validations for scDEAL. First, we added ten replications to all comparisons in the model ablation test (Figure 2B-D) using the same parameters to demonstrate model robustness (also suggested by Reviewer #2). Second, we revised the Cisplatin case study of Data 1 with strengthened evidence regarding predicted critical genes and pathway enrichment to validate the results with the help of a Cisplatin expert (Dr. Qi-en Wang). Details can be found in response to Comment 3.

Regarding the computational method in this study, we believe that we now have significant methodological innovations, especially after adapting the valuable comments from you and Reviewer #2. Although the basic elements of deep learning, such as transfer learning, are not new, our development represents the first-of-its-kind deep transfer learning model to predict drug response at the single-cell level by learning drug-gene relations in the bulk databases. This is highly non-trivial, as we are not merely adapting the existing model to this specific single-cell problem, but assembled a novel framework by targeting the specific problem and adding special adjustments, e.g., cell type regularization to maintain single-cell heterogeneity. Hence, this is not achievable by simply plugging the data into an existing deep learning model. This new framework utilized the data comprehensively and systematically, which significantly added values for drug-response prediction. In contrast, traditional differential analysis based on gene expression may be biased to cell types rather than drug response. We also designed a new integrated gradient (IG) score to directly retrieve marker genes that contribute to the prediction of drug response based on the neural network mechanism. Our results show that marker genes determined by the differential analysis using IG scores are directly related to drug response. Hope the above clarifications make sense to you.

2. The uniqueness of single-cell data is in providing information about cell population heterogeneity and predicting cell clusters/subpopulations. The authors show sample-specific response (treated/untreated) but fail to demonstrate that scDEAL predicts cluster-specific response. Reviewer's 2 Point 2.6 is poorly addressed.

Response: Thanks for your suggestion. We have now revised the Cisplatin case study to showcase the cluster-specific response in Figure 4 and Supplementary Data S1-2. Additionally, for Point 2.6, Reviewer 2 suggested adding more details regarding training data selection for model improvement. To answer this question, we explained that scDEAL only relies on bulk data for model training, and showcased that the performance could be enhanced by integrating additional bulk drug databases (Figure 2C). We discussed that more bulk data could be integrated into scDEAL in the future, while more scRNA-seq datasets with well-labeled drug responses could also help refine the parameter selection for scDEAL. In this round of review, Reviewer 2 was

satisfied with our response to Point 2.6. Meanwhile, we are glad to revise the content accordingly if you have additional suggestions regarding this point.

3-1. Reviewer's 1 Point 1.4 is not completely addressed. One example is "the carbon metabolism pathway to be a Cisplatin-sensitive target". I am not sure that I am aware of this pathway. Shouldn't it be "one-carbon"? Overall, the mentioned critical genes are quite superficially connected to cisplatin sensitivity and can't be considered as a validation of the approach.

3-2. Not clear what "a drug-sensitive marker to Cisplatin response" really means, and why BRD4 belongs to this category. There is no supporting evidence in the cited literature:

Yosinski, J., Clune, J., Bengio, Y. & Lipson, H. How transferable are features in deep neural networks? arXiv preprint arXiv:1411.1792 (2014)

Similar, what are the "mediators to induce Cisplatin resistance"?

3-3. The validated factors are positioned at the 126th and 56th places. Are the top TFs all false positive? Why don't they appear?

Response: We appreciate the Reviewer's carefulness in pointing out these issues. As the three questions are all related to the Data 1 case study, we answer them together below.

(i) In this revision, we invited Dr. Qi-En Wang (<https://cancer.osu.edu/find-a-researcher/search-researcher-directory/qi-en-wang>) to redesign the Cisplatin case study and interpret the prediction results. He is an associate professor at the Ohio State University with research topics focused on Cisplatin resistance in ovarian cancer. First, we found out that the two drug-holiday samples in the original study should be a mixture of sensitive and resistant cells. Since our model involves the regularization of cell clusters to polarize drug response predictions, the cell clusters from two drug-holiday samples may negatively influence the prediction of drug response as well as critical genes corresponding to drug response. To this end, we removed the two drug-holiday samples in the HN120 dataset and observed an increased F1-score for drug response prediction compared to the original labels. To validate the connection between critical genes and Cisplatin response, we specifically compared the HN120P (sensitive cells from the primary site) with HN120PCR (resistant cells from the primary site after a four-month Cisplatin treatment). We updated our findings in the manuscript accordingly.

(ii) According to Dr. Qi-en Wang's comment, we believe that predicting TFs from critical genes using third-party tools may not be appropriate to identify TFs contribute to drug responses accurately. We would like to point out that the previous TF results are based on a third-party software LISA, without statistical correlations to drug responses. LISA does not predict *cis*-regulatory elements but probes the effects of deleting putative regulatory TR cistromes on the chromatin regulatory potential model. It indeed includes false positives due to the in-silico perturbation mechanism. As suggested by Dr. Wang, discussing drug-related TFs purely from gene expression profiles maintains high risks and false positives. Therefore, we now decided to

enhance the functional interpretation by pathway enrichment analysis (highly related to DNA repair), and not to include TF analysis in our revision. We will derive additional efforts in discovering the TF-drug relations in our future studies.

4. Data on the OSCC model is supposed to be presented in Figures 3a, 3B and 3C but lacking.

Response: We have generated Figures for OSCC data similar to Figures 3A-C. The new figures can be found in Fig. 4A and Supplementary Figure S8.

5. It is unclear why scDEAL drug response prediction should correlate with pseudotime analysis. Also, there is no explanation of Data 6 samples in order to assess robustness of the experimental design.

Response: We considered the change of drug response to be a process from sensitive to resistant, e.g., remaining cells are becoming more resistant to a drug as the treatment period last. The drug response development, therefore, should be correlated with cell development at different treatment time points. The pseudotime analysis result can reflect cell development at the single-cell level. Also, we have proved that the drug response prediction of scDEAL wellly matched with the drug response defined in the original papers. Therefore, we hypothesize that the accurate prediction of drug response should be correlated with cell development in pseudotime. There are also papers that correlate pseudotime analysis with drug-response at the single-cell level, e.g., <https://doi.org/10.1038/s41598-021-97887-z>.

Two experiments were carried out to assess the robustness of scDEAL. Firstly, we performed robustness tests on all six scRNA-seq datasets via 80% stratified sub-sampling in terms of F1 score (Figure 1F). Secondly, also as suggested by Reviewer #2, we performed replication tests on comparisons regarding the model ablation test (Figures 1B-D). Both results showed relatively small variations of scDEAL on different single-cell datasets, supporting the robustness of the experimental design (see details in the response to Comment 1 from Reviewer #2).

Reviewer #2:

We thank the authors for their thorough and thoughtful revision. Overall, the revised manuscript has addressed all the comments we previously raised. In particular, we are pleased that the authors significantly extended and improved the method description (with few key updates on the method itself), the analyses, and the presentation of the results. We ask that a few minor points be addressed before the manuscript is accepted for publication:

1. - In Figure 2, the authors have added a robustness test for the performance of the model across different scRNA-seq datasets. However, the authors should also add confidence intervals (or, at least, show replicates) for all the other bar plots in Figure 2, as this would further demonstrate the robustness of the obtained results.

Response: Thanks for your advice. For experiments in Figures 2B-D, we performed ten

replications using the same parameters. The bar plots showed average F1 scores in each experiment, and error bars were added. As a result, scDEAL showed the highest average F1 scores of ten replicates in most of the comparisons except for Data 3, where only use GDSC database showed a higher average F1 score. On the other hand, the average standard deviation of the ten replicates in all six datasets using scDEAL is 0.06, smaller than the model without transfer learning (0.08), only use GDSC database (0.10), only use CCLE database (0.07), use autoencoder without cell type regularization (0.10), and use denoising autoencoder without cell type regularization (0.08).

2. - The addition of the cell cluster loss is a valuable improvement compared to the previous version of the method. The authors already showed that it improves performance across datasets compared to a baseline (Fig. 2D). We ask that the authors also show how it can better preserve the heterogeneity of scRNA-seq data, as the authors claim multiple times in the manuscript. For example, the authors could show latent representations obtained with/without this additional loss.

Response: Thanks for your suggestions. We added Fig. 2E and additional Supplementary Figure S5 to address your comment. Specifically, we showcase cells with cell cluster and drug response annotations using latent representations from scDEAL, with/without the cell type regularizer. The UMAP results showed that, after applying the cell type regularizer, the cells become more ordered and compact within a cluster.

Reviewer #3 (Remarks to the Author):

I thank the authors for providing satisfactory answers to the questions I posed.

We thank the endorsements from Reviewer 3.

REVIEWER COMMENTS

Reviewer #4 (Remarks to the Author): Expert in single-cell RNA-seq and cisplatin therapy

In this work the authors present an innovative and reliable framework to predict responsiveness of a specific drug at single cell level by leveraging information of extensive bulk RNA dataset for which the information of the responsiveness is available.

The authors addressed most of the points raised by Reviewer 1 during the second round of revisions. However, we would like to point out two aspects that in our opinion need to be clarified before the final acceptance of this work.

First, reviewer 1 asked to provide information about cell clusters/heterogeneity, which is one of the strengths of scRNAseq data. We agree with the authors that the introduction of the cell-type regularisation maintains the scRNAseq heterogeneity information. Nevertheless, it would be nice to know where the authors took the cell clusters info. From the codes it seems that a novel Louvain clustering approach from Scanpy was performed. However, the materials and methods section does not contain any parameters / functions of this framework. The authors should add this infos for reproducibility. Furthermore, It has been widely demonstrated that batch effects could bias the final clustering on scRNAseq data, the authors should either test batch effects on their new clustering or mention this as limitation of the study.

Second, we have found that the authors partially answered the points 3-1 and 3-2 raised in the second round of revision by reviewer 1 and regarding Cisplatin activity. Moreover, we have to point out that we have some concerns on the use of scRNAseq Data 1 and Data 2 from Sharma et al., 2018 - Nat Comm. These data, which cover a relevant part of the validation of the framework, have been recently criticised due to a potential sample swap: in particular, the genotype identity of HN120P (primary/sensitive line) is inconsistent with the one of the other HN120 cell lines (not primary cell line: M, CR , CRDH) but it resembles the one of HN137 not primary cell lines. The same has been identified for HN137P whose genotype identity is more similar to the one of the HN120 not primary cell lines. This suggests a possible switch that involves HN137P and HN120P (Ramazzotti et al., 2022 - Nat Comm). We are aware that this shouldn't affect the prediction analysis due to the fact that the swapped samples belong both to sensitive cell lines, and in a sense highlights a certain reliability of the approach. However, this may lead to potential misleading results, in particular, on the identification of top-sensitive and top-resistant CGs that are shown in Fig.3B and Fig.3C since the gene signature that the authors highlight is from HN120P. We please ask the authors either to re-run the analysis using the right cell line or to mention this inconsistency in the text. Finally, I am wondering if this could also be the reason for the mismatching of the response labels in an entire cluster when using Data 2 (Sup. Fig. S3).

In addition to what we have raised above, we have other minor concerns that could be relevant for the publication of this paper.

I) In several occasions (lines 111-112, line 136, etc.) the latent features in the bottleneck layer of the DAE are referred to as “selected” features, and the feature extraction process as a “feature selection”. This is not correct and potentially misleading, as autoencoders are not feature selection algorithms, rather dimensionality reduction methods. Indeed, AEs extract informative feature vectors of reduced dimensionality via a (non linear) combination of potentially all input variables. Of course, weights close to zero can nullify the relevance of any input feature in the latent representation, but as this is not guaranteed nor induced in the DAE implementation as described in the paper, the authors should change the wording to avoid confusion in the reader.

II) Data 6 (Bell et al.,) are from a mouse model, it is nice how scDEAL effectively deals with data derived from a different species even when the bulk data used to train the model are all from human cell lines. In my opinion it would be worth discussing this trans-species reliability on a few lines of the manuscript.

III) Supplementary Fig. S5 has panels with a scarce image resolution that makes it difficult to read the panels.

IV) Fig. 3C, I would ask the authors to change the colour of the dots. These can create confusion since the reader could associate them to the ones sensitive and resistant to Fig. 3A.

V) Along the manuscript and on the Figures, genes need to be written in italics.

VI) Pag 17 403-404 legend of the figure: DEG scores have to be listed before the Correlation as it is shown in the figure.

VII) Pag. 24 Line 658 Supplementary Fig. S6 should be Supplementary Fig. S4.

Reviewer #4 (Remarks to the Author): Expert in single-cell RNA-seq and cisplatin therapy

In this work the authors present an innovative and reliable framework to predict responsiveness of a specific drug at single cell level by leveraging information of extensive bulk RNA dataset for which the information of the responsiveness is available.

The authors addressed most of the points raised by Reviewer 1 during the second round of revisions. However, we would like to point out two aspects that in our opinion need to be clarified before the final acceptance of this work.

First, reviewer 1 asked to provide information about cell clusters/heterogeneity, which is one of the strengths of scRNAseq data. We agree with the authors that the introduction of the cell-type regularisation maintains the scRNAseq heterogeneity information. Nevertheless, it would be nice to know where the authors took the cell clusters info. From the codes it seems that a novel Louvain clustering approach from Scanpy was performed. However, the materials and methods section does not contain any parameters / functions of this framework. The authors should add this infos for reproducibility. Furthermore, It has been widely demonstrated that batch effects could bias the final clustering on scRNAseq data, the authors should either test batch effects on their new clustering or mention this as limitation of the study.

Response: We apologize for the misleading description. We did not develop a novel Louvain clustering approach in scDEAL. Instead, we used the open-source Louvain clustering code implemented in the igraph R package [1]. Since this approach has been widely used in other tools, such as our in-house scGNN [2], we believe it is unnecessary to demonstrate the reproducibility or batch effect correction in this work. We clearly mentioned the above information in the Methods section.

[1] Csardi, Gabor, and Tamas Nepusz. "The igraph software package for complex network research." *InterJournal, complex systems* 1695.5 (2006): 1-9.

[2] Wang, Juexin, et al. "scGNN is a novel graph neural network framework for single-cell RNA-Seq analyses." *Nature communications* 12.1 (2021): 1-11.

Second, we have found that the authors partially answered the points 3-1 and 3-2 raised in the second round of revision by reviewer 1 and regarding Cisplatin activity. Moreover, we have to point out that we have some concerns on the use of scRNAseq Data 1 and Data 2 from Sharma et al., 2018 - Nat Comm. These data, which cover a relevant part of the validation of the framework, have been recently criticised due to a potential sample swap: in particular, the genotype identity of HN120P (primary/sensitive line) is inconsistent with the one of the other HN120 cell lines (not primary cell line: M, CR , CRDH) but it

resembles the one of HN137 not primary cell lines. The same has been identified for HN137P whose genotype identity is more similar to the one of the HN120 not primary cell lines. This suggests a possible switch that involves HN137P and HN120P (Ramazzotti et al., 2022 - Nat Comm). We are aware that this shouldn't affect the prediction analysis due to the fact that the swapped samples belong both to sensitive cell lines, and in a sense highlights a certain reliability of the approach. However, this may lead to potential misleading results, in particular, on the identification of top-sensitive and top-resistant CGs that are shown in Fig.3B and Fig.3C since the gene signature that the authors highlight is from HN120P. We please ask the authors either to re-run the analysis using the right cell line or to mention this inconsistency in the text. Finally, I am wondering if this could also be the reason for the mismatching of the response labels in an entire cluster when using Data 2 (Sup. Fig. S3).

Response: We appreciated the Reviewer providing us such helpful information. We re-run the analysis by reassembling the HN120 and HN137 datasets as suggested. Specifically, we reassembled a new HN120 data with cells originally labeled as HN137P, HN120PCR, HN120M, and HN120MCR, and a new HN137 data with cells originally labeled as HN120P, HN120PCR, HN120M, and HN120MCR. A similar grid-search method was applied to optimize the reassembled data analysis using scDEAL. As a result, we found that:

(i) The new drug response prediction F1-score of re-assembled HN120 data is 0.750 (originally 0.839) and of re-assembled HN137 data is 0.764 (originally 0.765), and all the remaining evaluation scores are slightly lower than the original result. The results showcased that, regardless of data assembly, scDEAL achieved competitive response prediction to Cisplatin. We fully agree with the Reviewer that such results highlight a certain reliability of the approach and strengthened the reliability of our approach. It also indicated the potential usage of scDEAL for combined data from different patients.

(ii) The critical genes identified in the re-assembled data between HN120P and HN120PCR aligned well with our previous results. All important genes related to Cisplatin resistance are found in both results. Such results indicated that scDEAL can find the real critical genes to drug response even though the sensitive and resistant tissues are derived from different patients.

(iii) In the original HN137 (Data 2), the Cisplatin response in HN137M cell group was completely mismatched. However, in the reassembled HN137 data, such a situation no longer exists. So that we believe such a data swap could be one of the reasons that caused the entire mismatch in this data.

Having all said that, we decide to discuss these new results in the Discussion section and

included them in the supplementary files (Supplementary S9, also showcased below for your convenience) rather than replace our previous results.

Supplementary Figure S9. scDEAL analysis of re-assembled HN120 and HN137 data. The scRNA-seq data was re-assembled by switching the HN120P with HN137P in the original datasets (marked with *). UMAPs with sample labels, ground truth drug response, and predicted drug response are showcased. For each analysis, the optimized (based on grid-search method) parameters and evaluation scores were given below.

In addition to what we have raised above, we have other minor concerns that could be relevant for the publication of this paper.

I) In several occasions (lines 111-112, line 136, etc.) the latent features in the bottleneck layer of the DAE are referred to as “selected” features, and the feature extraction process as a “feature selection”. This is not correct and potentially misleading, as autoencoders are not feature selection algorithms, rather dimensionality reduction methods. Indeed, AEs extract informative feature vectors of reduced dimensionality via a (non linear) combination of potentially all input variables. Of course, weights close to zero can nullify the relevance of any input feature in the latent representation, but as this is not guaranteed nor induced in the DAE implementation as described in the paper, the authors should change the wording to avoid confusion in the reader.

Response: Thanks for pointing out this. We changed “selected features” as “low-dimension features” and “feature selection” as “feature reduction”.

II) Data 6 (Bell et al.,) are from a mouse model, it is nice how scDEAL effectively deals with data derived from a different species even when the bulk data used to train the model

are all from human cell lines. In my opinion it would be worth discussing this trans-species reliability on a few lines of the manuscript.

Response: We appreciate Reviewer's suggestion. We are also aware that the cross-species reliability is a very interesting and important task for drug prediction. However, due to the limited benchmark data in the public domain, we could not make a solid conclusion nor optimize our framework for tackling the cross-species task. We have added a paragraph in Discussion regarding this and will carefully consider it in our future study.

III) Supplementary Fig. S5 has panels with a scarce image resolution that makes it difficult to read the panels.

Response: We have enhanced the image resolution in Fig. S5 (now Fig. S4). Since Data 5 and 6 were already showcased in Fig 2E, we removed them from this Supplementary Figure.

IV) Fig. 3C, I would ask the authors to change the colour of the dots. These can create confusion since the reader could associate them to the ones sensitive and resistant to Fig. 3A.

Response: Thanks for the suggestion. Instead of changing colors, we added figure legends to clearly state the cell cluster labels are from the predicted results, not from Fig. 3A.

V) Along the manuscript and on the Figures, genes need to be written in italics.

Response: Thanks for the comment. We have made all genes in italics.

VI) Pag 17 403-404 legend of the figure: DEG scores have to be listed before the Correlation as it is shown in the figure.

Response: We have fixed this.

VII) Pag. 24 Line 658 Supplementary Fig. S6 should be Supplementary Fig. S4.

Response: We have fixed this.

REVIEWERS' COMMENTS

Reviewer #5 (Remarks to the Author): Expert in statistics, machine learning, bioinformatics and genomics; co-reviewed with Reviewer #4

We reviewed the authors' responses and the revised version of the paper.

We think that their replies satisfy our main concerns.

In particular, we thank the authors for the explanation about the clustering method and its addition to the materials and methods section.

Regarding the sample swap we appreciated that the authors re-performed the analyses with the correct samples and that, as expected, the response prediction score F1 was comparable to the previous results as well as the list of critical genes.

We would like to point out that the authors should double check the text they added to the paper regarding the datasets 1 and 2 reassembly, since it likely contains some typos/repetition (e.g. HN120PCR, HN120M, and HN120MCR are written two times). As a consequence, for the sake of completeness, we suggest checking also the samples that were added to the re-assembled datasets: the UMAP representation of HN120P* looks strange. However, we believe that this is not going to change the main results / message of the study, which is strongly supported by several successful experiments on various datasets.

In conclusion, the authors exhaustively answered all our inquiries, and we believe that their work at this stage constitutes a relevant novel contribution to the field.

Reviewer #5 (co-reviewed with **Reviewer #4**): Expert in statistics, machine learning, bioinformatics and genomics.

We would like to point out that the authors should double-check the text they added to the paper regarding the datasets 1 and 2 reassembly, since it likely contains some typos/repetition (e.g., HN120PCR, HN120M, and HN120MCR are written two times). As a consequence, for the sake of completeness, we suggest checking also the samples that were added to the re-assembled datasets: the UMAP representation of HN120P* looks strange. However, we believe that this is not going to change the main results/message of the study, which is strongly supported by several successful experiments on various datasets.

Response: We immensely appreciate the Reviewer's endorsement and are glad to know our previous revision satisfied the reviewers. For the two remaining comments:

- (1) we apologize for the typo. It should be "a new HN137 data with cells originally labeled as HN120P, HN137PCR, HN137M, and HN137MCR". We have corrected it in the manuscript.
- (2) The separation of the HN120P* cluster (originally HN137P) is due to the existence of paired-end and single-end data in the original study. As you can see in Daniele Ramazzotti's paper [1] (see the screenshot of Figure 1B below), the paired-end HN137P cells were grouped in Cluster 1, and single-end HN137P cells were grouped in Cluster 2. Additionally, the single-end HN137P cells were also separated into two subclusters. In our experiment, we did not distinguish the paired-end and single-cell HN137P cells; thus, it is reasonable to have three clusters in Supplementary Figure 9.

[1] Ramazzotti, D. et al. Variant calling from scRNA-seq data allows the assessment of cellular identity in patient-derived cell lines. *Nature Communications* **13**, 2718, doi:10.1038/s41467-022-30230-w (2022)

The screenshot of Figure 1B from Daniele Ramazzotti's paper. Paired-end and single-end HN137P cells are separated in Clusters 1 and 2, respectively, and the single-end HN137P cells were also separated into two subclusters.